# Impacts of aerosol-photolysis interaction and aerosol-radiation feedback on surface-layer ozone in North China during multi-pollutant air pollution episodes

Hao Yang[1], Lei Chen[1], Hong Liao[1], Jia Zhu[1], Wenjie Wang[2], Xin Li[2]

[1]Jiangsu Key Laboratory of Atmospheric Environment Monitoring and Pollution Control, Jiangsu Collaborative Innovation Center of Atmospheric Environment and Equipment Technology, School of Environmental Science and Engineering, Nanjing University of Information Science & Technology, Nanjing 210044, China
[2]State Joint Key Laboratory of Environmental Simulation and Pollution Control, College of Environmental Sciences and Engineering, Peking University, Beijing 100871, China

**Correspondence:** Lei Chen (chenlei@nuist.edu.cn) and Hong Liao (hongliao@nuist.edu.cn)

## Abstract

We examined the impacts of aerosol-radiation interactions, including the effects of aerosol-photolysis interaction (API) and aerosol-radiation feedback (ARF), on surface-layer ozone ($O_3$) concentrations during four multi-pollutant air pollution episodes characterized by high $O_3$ and $PM_{2.5}$ levels during 28 July to 3 August 2014 (Episode1), 8-13 July 2015 (Episode2), 5-11 June 2016 (Episode3), and 28 June to 3 July 2017 (Episode4) in North China, by using the Weather Research and Forecasting with Chemistry (WRF-Chem) model embedded with an integrated process analysis scheme. Our results show that API and ARF reduced the daytime shortwave radiative fluxes at the surface by 92.4~102.9 W $m^{-2}$ and increased daytime shortwave radiative fluxes in the atmosphere by 72.8~85.2 W $m^{-2}$, as the values were averaged over the complex air pollution areas (CAPAs) in each of the four episodes. As a result, the stabilized atmosphere decreased the daytime planetary boundary layer height and 10 m wind speed by 129.0~249.0 m and 0.05~0.15 m $s^{-1}$, respectively, in CAPAs in the four episodes. Aerosols were simulated to reduce the daytime near-surface photolysis rates of $J[NO_2]$ and $J[O^1D]$ by $1.8 \times 10^{-3}$~$2.0 \times 10^{-3}$ $s^{-1}$ and $5.7 \times 10^{-6}$~$6.4 \times 10^{-6}$ $s^{-1}$, respectively, in CAPAs in the four episodes. All the four episodes show the same conclusion that the reduction in $O_3$ by API is larger than that by ARF. API (ARF) was simulated to change daytime surface-layer $O_3$ concentrations by -8.5 ppb (-2.9 ppb), -10.3 ppb (-1.0 ppb), -9.1 ppb (-0.9 ppb) and -11.4 ppb (+0.7 ppb) in CAPAs of the four episodes, respectively. Process analysis indicated that the weakened $O_3$ chemical production made the greatest contribution to API effect, while the reduced vertical mixing was the key process for ARF effect. Our conclusions suggest that future $PM_{2.5}$ reductions may lead to $O_3$ increases due to the weakened aerosol-radiation interactions, which should be considered in air quality planning.

## 1 Introduction

The characteristics of air pollution in China during recent years are changing from the single pollutant (e.g., $PM_{2.5}$, particulate matter with an aerodynamic equivalent diameter of 2.5 μm or less) to multiple pollutants (e.g., $PM_{2.5}$ and ozone ($O_3$)) (Zhao et al., 2018; Zhu et al., 2019), and the synchronous occurrence of high $PM_{2.5}$ and $O_3$ concentrations has been frequently observed, especially during the warm seasons (Dai et al., 2021; Qin et al., 2021). Qin et al. (2021) reported that the co-occurrence of $PM_{2.5}$ and $O_3$ pollution days (days with $PM_{2.5}$ concentration > 75 μg $m^{-3}$ as well as maximum daily 8 h average ozone concentration > 80 ppb) exceeded 324 days in eastern China during 2015-2019. Understanding the complex air pollution is essential for making plans to improve air quality in China.

Aerosols can influence $O_3$ by changing meteorology through absorbing and scattering solar radiation (defined as aerosol-radiation feedback (ARF) in this work) (Albrecht et al., 1989; Haywood et al., 2000; Lohmann et al., 2005), which influences air quality by altering the chemical reactions, transport and deposition of the pollutant (Gao et al., 2018; Qu et al., 2021; Xing et al., 2017; Zhang et al., 2018). Many studies examined the feedback between aerosols and meteorology (Gao et al., 2015; Gao et al., 2016a; Qiu et al., 2017; Chen et al., 2019; Zhu et al., 2021). For example, Gao et al. (2015) used the WRF-Chem model to investigate the feedbacks between aerosols and meteorological variables over the North China Plain in January 2013, and pointed out that aerosols caused a decrease in surface temperature by 0.8-2.8 °C but an increase of 0.1-0.5 °C around 925 hPa. By using the same WRF-Chem model, Qiu et al. (2017) reported that the surface downward shortwave radiation and PBLH were reduced by 54.6 W $m^{-2}$ and 111.4 m, respectively, due to aerosol direct radiative effect during 21-27 February 2014 in the North China Plain. Such aerosol-induced changes in meteorological fields are expected to influence $O_3$ concentrations during multi-pollutant episodes with high concentrations of air pollutants.

Aerosols can also influence $O_3$ by altering photolysis rates (defined as aerosol-photolysis interaction (API) in this work) (Dickerson et al., 1997; Liao et al., 1999; Li

et al., 2011; Lou et al., 2014). Dickerson et al. (1997) reported that the presence of
pure scattering aerosol increased ground level ozone in the eastern United States by
20 to 45 ppb, while the presence of strongly absorbing aerosol reduced ground level
ozone by up to 24 ppb. Wang et al. (2019) found that aerosols reduced the net ozone
production rate by 25% by reducing the photolysis frequencies during a
comprehensive field observation in Beijing in August 2012. Such aerosol-induced
changes in photolysis rates are expected to influence $O_3$ concentrations during multi-
pollutant episodes with high concentrations of air pollutants.
Few previous studies quantified the effects of ARF and API on $O_3$ concentrations.
Xing et al. (2017) applied a two-way online coupled WRF-CMAQ model and
reported that the combination of API and ARF reduced the surface daily maximum 1
h $O_3$ (MDA1 $O_3$) by up to 39 $\mu g\ m^{-3}$ over China during January 2013. Qu et al. (2021)
found, by using the UK Earth System Model (UKESM1), that ARF reduced the
annual average surface $O_3$ by 3.84 ppb (14.9%) in the North China Plain during 2014.
Gao et al. (2020) analyzed the impacts of API on $O_3$ by using the WRF-Chem model
and reported that API reduced surface $O_3$ by 10.6 ppb (19.0%), 8.6 ppb (19.4%), and
8.2 ppb (17.7%) in Beijing, Tianjin, and Shijiazhuang, respectively, during October
2018. However, these previous studies mostly examined either ARF or API and did
not examine their total and respective roles in $O_3$ pollution in China. Furthermore,
these previous studies lacked process understanding about the impacts of ARF and
API on $O_3$ pollution under co-occurrence of $PM_{2.5}$ and $O_3$ pollution events.
The present study aims to quantify the respective/combined impacts of ARF and
API on surface $O_3$ concentrations by using the WRF-Chem model, and to identify the
prominent physical and/or chemical processes responsible for ARF and API effects by
using an integrated process rate (IPR) analysis embedded in the WRF-Chem model.
We carry on simulations and analyses on four multi-pollutant air pollution episodes
(Episode1: 28 July to 3 August 2014; Episode2: 8-13 July 2015; Episode3: 5-11 June
2016; Episode4, 28 June to 3 July 2017) in North China with high $O_3$ and $PM_{2.5}$ levels
(the daily mean $PM_{2.5}$ and the maximum daily 8-h average $O_3$ concentration are larger
than 75 $\mu g\ m^{-3}$ and 80 ppb, respectively). These episodes are selected because (1)

these events with high concentrations of both $PM_{2.5}$ and $O_3$ are the major subjects of air pollution control, (2) high concentrations of both $PM_{2.5}$ and $O_3$ allow one to obtain the strongest signals of ARF and API, (3) the measurements of $J[NO_2]$ during 2014 and 2015 from Peking University site (Wang et al., 2019) can help to constrain the simulated photolysis rates of $NO_2$, and (4) selected events cover different years of 2014 to 2017 during which the governmental Air Pollution Prevention and Control Action Plan was implemented (the changes in emissions and observed $PM_{2.5}$ in the studied region during 2014-2017 are shown in Fig. S1). We expect that the conclusions obtained from multiple episodes represent the general understanding of the impacts of ARF and API.

The model configuration, numerical experiments, observational data, and the integrated process rate analysis are described in section 2. Section 3 shows the model evaluation. Results and discussions are presented in section 4, and the conclusions are summarized in section 5.

## 2 Methods

### 2.1 Model configuration

The version 3.7.1 of the online-coupled Weather Research and Forecasting with Chemistry (WRF-Chem) model (Grell et al., 2005; Skamarock et al., 2008) is used in this study to explore the impacts of aerosol-radiation interactions on surface-layer $O_3$ in North China. WRF-Chem can simulate gas phase species and aerosols coupled with meteorological fields, and has been widely used to investigate air pollution over North China (Gao et al., 2016a; Gao et al., 2020; Wu et al., 2020). As shown in Fig. 1, we design two nested model domains with the number of grid points of 57 (west–east) $\times$ 41 (south–north) and 37 (west–east) $\times$ 43 (south–north) at 27 and 9 km horizontal resolutions, respectively. The parent domain centers at (39 °N, 117 °E). The model contains 29 vertical levels from the surface to 50 hPa, with 14 levels below 2 km for the fully description of the vertical structure of planetary boundary layer (PBL).

The Carbon Bond Mechanism Z (CBM-Z) is selected as the gas-phase chemical mechanism (Zaveri and Peters, 1999), and the full 8-bin MOSAIC (Model for

Simulating Aerosol Interactions and Chemistry) aerosol module with aqueous chemistry is used to simulate aerosol evolution (Zaveri et al., 2008). The photolysis rates are calculated by the Fast-J scheme (Wild et al., 2000). Other major physical parameterizations used in this study are listed in Table 1.

The initial and boundary meteorological conditions are provided by the National Centers for Environmental Prediction (NCEP) Final Analysis data with a spatial resolution of $1° \times 1°$. In order to limit the model bias of simulated meteorological fields, the four-dimensional data assimilation (FDDA) is used with the nudging coefficient of $3.0 \times 10^{-4}$ for wind, temperature and humidity (no analysis nudging is applied for the inner domain) (Lo et al., 2008; Otte, 2008). Chemical initial and boundary conditions are obtained from the Model for Ozone and Related chemical Tracers, version 4 (MOZART-4) forecasts (Emmons et al., 2010).

Anthropogenic emissions in these four episodes are taken from the Multi-resolution Emission Inventory for China (MEIC) (http://www.meicmodel.org/) (Li et al., 2017a). These emission inventories provide emissions of sulfur dioxide ($SO_2$), nitrogen oxides ($NO_x$), carbon monoxide (CO), non-methane volatile organic compounds (NMVOCs), carbon dioxide ($CO_2$), ammonia ($NH_3$), black carbon (BC), organic carbon (OC), $PM_{10}$ (particulate matter with aerodynamic diameter is 10 μm and less) and $PM_{2.5}$. Emissions are aggregated from four sectors, including power generation, industry, residential, and transportation, with $0.25° \times 0.25°$ spatial resolution. Biogenic emissions are calculated online by the Model of Emissions of Gases and Aerosols from Nature (MEGAN) (Guenther et al., 2006).

**2.2 Numerical experiments**

To quantify the impacts of API and ARF on $O_3$, three experiments have been conducted: (1) BASE – the base simulation coupled with the interactions between aerosol and radiation, which includes both impacts of API and ARF; (2) NOAPI – the same as the BASE case, but the impact of API is turned off ( aerosol optical properties are set to zero in the photolysis module), following Wu et al. (2020); (3) NOALL – both the impacts of API and ARF are turned off (removing the mass of aerosol species

when calculating aerosol optical properties in the optical module), following Qiu et al. (2017). The differences between BASE and NOAPI (i.e., BASE minus NOAPI) represent the impacts of API. The contributions from ARF can be obtained by comparing NOAPI and NOALL (i.e., NOAPI minus NOALL). The combined effects of API and ARF on $O_3$ concentrations can be quantitatively evaluated by the differences between BASE and NOALL (i.e., BASE minus NOALL).

All the experiments in Episode1, Episode2, Episode3 and Episode4 are conducted from 26 July to 3 August 2014, 6-13 July 2015, 3-11 June 2016, and 26 June to 3 July 2017, respectively, with the first 40 hours as the model spin-up in each case. Simulation results from the BASE cases of the four episodes are used to evaluate the model performance.

**2.3 Observational data**

Simulation results are compared with meteorological and chemical measurements. The surface-layer meteorological data (2 m temperature ($T_2$), 2 m relative humidity ($RH_2$), and 10 m wind speed ($WS_{10}$)) with the temporal resolution of 3 h at ten stations (Table S1) are obtained from NOAA's National Climatic Data Center (https://gis.ncdc.noaa.gov/maps/ncei/cdo/hourly). The radiosonde data of temperature at 08:00 and 20:00 LST in Beijing (39.93 °N, 116.28 °E) are provided by the University of Wyoming (http://weather.uwyo.edu/). Observed hourly concentrations of $PM_{2.5}$ and $O_3$ at thirty-two sites (Table S2) in North China are collected from the China National Environmental Monitoring Center (CNEMC). The photolysis rate of nitrogen dioxide (J[$NO_2$]) measured at the Peking University site (39.99 °N, 116.31 °E) is also used to evaluate the model performance. More details about the measurement technique of J[$NO_2$] can be found in Wang et al. (2019). The aerosol optical depth (AOD) at Beijing site (39.98°N, 116.38°E) is provided by AERONET (level 2.0, http://aeronet.gsfc.nasa.gov/). The AOD at 675 nm and 440 nm are used to derive the AOD at 550 nm to compare with the simulated ones.

**2.4 Integrated process rate analysis**

Integrated process rate (IPR) analysis has been widely used to quantify the

contributions of different processes to $O_3$ variations (Goncalves et al., 2009; Gao et al.,
2016b; Tang et al., 2017; Gao et al., 2018). In this study, four physical/chemical
processes are considered, including vertical mixing (VMIX), net chemical production
(CHEM), horizontal advection (ADVH), and vertical advection (ADVZ). VMIX is
initiated by turbulent process and closely related to PBL development, which
influences $O_3$ vertical gradients. CHEM represents the net $O_3$ chemical production
(chemical production minus chemical consumption). ADVH and ADVZ represent
transport by winds (Gao et al., 2016b). In this study, we define ADV as the sum of
ADVH and ADVZ.

## 3 Model evaluation

Reasonable representation of observed meteorological and chemical variables by
the WRF-Chem model can provide foundation for evaluating the impacts of aerosols
on surface-layer ozone concentrations. The model results presented in this section are
taken from the BASE cases in the four episodes. The concentrations of air pollutants
are averaged over the thirty-two observation sites in Beijing, Tianjin and Baoding. To
ensure the data quality, the mean value for each time is calculated only when
concentrations are available at more than sixteen sites, as done in Li et al. (2019a).

### 3.1 Chemical simulations

Figure 2 shows the temporal variations of observed and simulated $PM_{2.5}$ and $O_3$
concentrations over North China for the four episodes. As shown in Fig. 2, the
temporal variations of observed $PM_{2.5}$ can be well performed by the model with index
of agreement (IOA) of 0.68, 0.68, 0.67 and 0.44 and normalized mean bias (NMB) of
-19.2%, 4.1%, 30.4% and 13.9% during Episode1, Episode2, Episode3 and Episode4,
respectively. The model also tracks well the diurnal variation of $O_3$ over the North
China, with IOA of 0.89, 0.94, 0.92 and 0.87 and NMB of -12.0%, -0.4%, 1.6% and -
13.8% for Episode1, Episode2, Episode3 and Episode4, respectively.
Figure S2 shows the correlation between observed and simulated AOD at 550
nm in Beijing. In the WRF-Chem model, the AOD at 550 nm are calculated by using
the values at 400 and 600 nm according to the Angstrom exponent. Analyzing Fig. S2,
the model can reproduce the observed AOD with R of 0.7 and NMB of 7.9%.

**3.2 Meteorological simulations**

Figure 3 shows the time series of observed and simulated $T_2$, $RH_2$, $WS_{10}$ and
$J[NO_2]$ during the four episodes. The observed $T_2$, $RH_2$, $WS_{10}$ are averaged over the
ten meteorological observation stations, and the $J[NO_2]$ are measured at Peking
University. Most of the monitored $J[NO_2]$ in Episode3 and Episode4 are unavailable,
so the comparison of $J[NO_2]$ in Episode3 and Episode4 is not shown. Generally, the
model can depict the temporal variations of $T_2$ fairly well with IOA of 0.94~0.98 and
the mean bias (MB) of -1.9~-0.6 °C. For $RH_2$, the IOA and MB are 0.90~0.98 and -
6.5%~1.9%, respectively. Although WRF-Chem model overestimates $WS_{10}$ with the
MB of 0.6~1.0 m s$^{-1}$, the IOA for $WS_{10}$ is 0.70~0.83 and the root-mean-square error
(RMSE) is 0.9~1.5 m s$^{-1}$, which is smaller than the threshold of model performance
criteria (2 m s$^{-1}$) proposed by Emery et al. (2001). The positive bias in wind speed can
also be reproduced in other studies (Zhang et al., 2010; Gao et al., 2015; Liao et al.,
2015; Qiu et al., 2017). The predicted $J[NO_2]$ agrees well with the observations with
IOA of 0.98~0.99 and NMB of 6.8%~6.9%. We also conduct comparisons of
observed and simulated temperature profiles at 08:00 and 20:00 LST in Beijing during
the four episodes (Fig. S3). The vertical profiles of observed temperature can be well
captured by the model in these four complex air pollution episodes. Generally, the
WRF-Chem model can reasonably reproduce the temporal variations of observed
meteorological parameters.

## 4 Results

We examine the impacts of aerosol-radiation interactions on $O_3$ concentrations
with a special focus on the complex air pollution areas (CAPAs, Fig. S4) in the four
episodes, where the daily mean simulated $PM_{2.5}$ and MDA8 (maximum daily 8-h
average) $O_3$ concentrations are larger than 75 µg m$^{-3}$ and 80 ppb, respectively, based
on the National Ambient Air Quality Standards (http://www.mee.gov.cn).

**4.1 Impacts of aerosol-radiation interactions on meteorology**

Figure 4 shows the impacts of aerosol-radiation interactions on shortwave
radiation at the surface (BOT_SW), shortwave radiation in the atmosphere
(ATM_SW), PBLH, and $WS_{10}$ during the daytime (08:00-17:00 LST) from Episode1
to Episode4. Analyzing the results of the interactions between aerosol and radiation
(the combined impacts of API and ARF), BOT_SW is decreased over the entire
simulated domain in the four episodes with the decreases of 93.2 W $m^{-2}$ (20.5%),
100.3 W $m^{-2}$ (19.5%), 92.4 W $m^{-2}$ (19.2%) and 102.9 W $m^{-2}$ (20.7%) over CAPAs,
respectively. Contrary to the changes in BOT_SW, ATM_SW is increased
significantly in the four episodes with the increases of 72.8 W $m^{-2}$ (25.3%), 85.2 W $m^{-2}$
(29.0%), 73.7 W $m^{-2}$ (26.4%) and 76.9 W $m^{-2}$ (25.8%) over CAPAs, respectively.
The decreased BOT_SW perturbs the near-surface energy flux, which weakens
convection and suppresses the development of PBL (Li et al., 2017b). The mean
PBLHs over CAPAs are decreased by 129.0 m (13.0%), 249.0 m (20.9%), 224.6 m
(19.0%) and 227.0 m (20.9%), respectively. $WS_{10}$ exhibits overall reductions over
CAPAs and is calculated to decrease by 0.12 m $s^{-1}$ (3.6%), 0.05 m $s^{-1}$ (1.6%), 0.12 m
$s^{-1}$ (3.0%) and 0.15 m $s^{-1}$ (4.3%), for the four episodes, respectively. We also examine
the changed meteorological variables caused by API and ARF respectively. As shown
in Fig. S5 and S6, API has little impact on meteorological variables; which means the
major contributor to the meteorology variability is ARF.
**4.2 Impacts of aerosol-radiation interactions on photolysis**
Figure 5 shows the spatial distributions of mean daytime surface-layer $PM_{2.5}$
concentrations simulated by BASE cases and the changes in $J[NO_2]$ and $J[O^1D]$ due
to aerosol-radiation interactions from Episode1 to Episode4. When the combined
impacts (API and ARF) are considered, $J[NO_2]$ and $J[O^1D]$ are decreased over the
entire domain in the four episodes, and the spatial patterns of changed $J[NO_2]$ and
$J[O^1D]$ are similar to that of simulated $PM_{2.5}$. Analyzing the four simulated episodes,
the surface $J[NO_2]$ averaged over CAPAs are decreased by $1.8 \times 10^{-3}$ $s^{-1}$ (40.5%), 2.0
$\times 10^{-3}$ $s^{-1}$ (36.8%), $1.8 \times 10^{-3}$ $s^{-1}$ (36.0%), and $2.0 \times 10^{-3}$ $s^{-1}$ (38.0%), respectively. The
decreased surface $J[O^1D]$ over CAPAs are $6.1 \times 10^{-6}$ $s^{-1}$ (48.8%), $6.3 \times 10^{-6}$ $s^{-1}$
(41.4%), $5.7 \times 10^{-6}$ $s^{-1}$ (44.6%), and $6.4 \times 10^{-6}$ $s^{-1}$ (46.9%), respectively. Figure S7
exhibits the impacts of API and ARF on surface $J[NO_2]$ and $J[O^1D]$. Conclusions can
be summarized that $J[NO_2]$ and $J[O^1D]$ are significantly modified by API and little
affected by ARF.

**4.3 Impacts of aerosol-radiation interactions on $O_3$**

Figure 6 shows the changes in surface-layer $O_3$ due to API, ARF, and the
combined effects (denoted as ALL) from Episode1 to Episode4. As shown in Fig.
6(a1-a4), API alone leads to overall surface $O_3$ decreases over the entire domain with
average reductions of 8.5 ppb (10.2%), 10.3 ppb (11.8%), 9.1 ppb (11.2%), and 11.4
ppb (12.2%) over CAPAs in the four episodes, respectively. The changes can be
explained by the substantially diminished UV radiation due to aerosol loading, which
significantly weakens the efficiency of photochemical reactions and restrains $O_3$
formation. However, the decreased surface $O_3$ concentrations due to ARF are only 2.9
ppb (3.2%, Fig. 6(b1)), 1.0 ppb (1.1%, Fig. 6(b2)) and 0.9 ppb (1.0%, Fig. 6(b3)) for
the Episode1 to Episode3 but ARF increased surface $O_3$ concentrations by 0.7 ppb
(0.5%, Fig.6(b4)) during Episode4, which was caused by the enhancement of
chemical production (Fig. S10 and Section 4.4). All the episodes show same
conclusion that the reduction in $O_3$ by API is larger than that by ARF. Fig. 6(c1-c4)
presents the combined effects of API and ARF. Generally, aerosol-radiation
interactions decrease the surface $O_3$ concentrations by 11.4 ppb (13.7%), 11.3 ppb
(13.0%), 10.0 ppb (12.3%) and 10.7 ppb (11.6%) averaged over CAPAs in the four
episodes, respectively.

**4.4 Influencing mechanism of aerosol-radiation interactions on $O_3$**

Figure 7a shows mean results of the four episodes (Episode1, Episode2,
Episode3 and Episode4) in diurnal variations of simulated daytime surface-layer $O_3$
concentrations from BASE, NOAPI and NOALL cases averaged over CAPAs. All the
experiments (BASE, NOAPI and NOALL) present $O_3$ increases from 08:00 LST. It is
shown that the simulated $O_3$ concentrations in BASE case increase more slowly than
that in NOAPI and NOALL cases. To explain the underlying mechanisms of API and
ARF impacts on $O_3$, we quantify the variations in contributions of different processes
(ADV, CHEM, and VMIX) to $O_3$ by using the IPR analysis.

Figure 7b shows hourly surface $O_3$ changes induced by each physical/chemical

process (i.e., ADV, CHEM, and VMIX) in BASE case averaged from Episode1 to
Episode4. The significant positive contribution to the hourly variation in $O_3$ is
contributed by VMIX, and the contribution reaches the maximum at about 09:00 LST.
Since VMIX increases the surface $O_3$ concentrations by transporting $O_3$ from aloft
(where $O_3$ concentrations are high) to the surface layer (Tang et al., 2017; Xing et al.,
2017; Gao et al., 2018). The CHEM process makes negative contributions at around
09:00 and 16:00 LST, which means that the chemical consumption of $O_3$ is stronger
than the chemical production. At noon, the net chemical contribution turns to be
positive due to stronger solar UV radiation. The contribution from all the processes
(NET, the sum of VMIX, CHEM, and ADV) to $O_3$ variation is peaked at the noon and
then becomes weakened. After sunset (17:00 LST), the NET contribution turns to be
negative over CAPAs, leading to $O_3$ decrease.

Figure 7c shows the changes in hourly process contributions caused by API

averaged from Episode1 to Episode4. The chemical production of $O_3$ is suppressed
significantly due to aerosol impacts on photolysis rates. The weakened $O_3$ chemical
production decreases the contribution from CHEM, and results in a negative value of
CHEM_DIF (-3.44 ppb $h^{-1}$). In contrast to CHEM_DIF, the contribution from
changed VMIX (VMIX_DIF) to $O_3$ concentration due to API is always positive, and
the mean value is +3.26 ppb $h^{-1}$. The positive change in VMIX due to API may be
associated with the different vertical gradient of $O_3$ between BASE and NOAPI cases
(Gao et al., 2020), as shown in Fig. 8a. The impact of API on ADV process is
relatively small (-0.26 ppb $h^{-1}$). NET_DIF, namely the sum of VMIX_DIF,
CHEM_DIF and ADV_DIF, indicates the differences in hourly $O_3$ changes caused by
API. As shown in Fig. 7c, NET_DIF is almost negative during the daytime over
CAPAs with the mean value of -0.44 ppb $h^{-1}$. This is because the decreases in CHEM
and ADV are larger than the increases in VMIX caused by API; the $O_3$ decrease is
mainly attributed to the significantly decreased contribution from CHEM. The
maximum difference in $O_3$ between BASE and NOAPI appears at 11:00 LST with a
value of -12.5 ppb (Fig. 7a).

Figure 7d shows the impacts of ARF on each physical/chemical process

contribution to the hourly $O_3$ variation averaged from Episode1 to Episode4. At 08:00
LST, the change in VMIX due to ARF is large with a value of -3.5 ppb $h^{-1}$, resulting in
a net negative variation with all processes considered. The decrease in $O_3$ reaches the
maximum with the value of 5.0 ppb at around 08:00 LST over CAPAs (Fig. 7a).
During 09:00 to 16:00 LST, the positive VMIX_DIF (mean value of +0.10 ppb $h^{-1}$) or
the positive CHEM_DIF (mean value of +0.75 ppb $h^{-1}$) is the major process to
positive NET_DIF. The positive VMIX_DIF is related to the evolution in boundary
layer during the daytime. The VOCs/$NO_x$ ratio is calculated to classify sensitivity
regimes and to indicate the possible $O_3$ responses to changes in VOCs and/or $NO_x$
concentrations. $O_3$ production is VOC-limited if the ratio is less than 4, and is $NO_x$-
limited if the ratio is larger than 15 (Edson et al., 2017; Li et al., 2017c). The ratio of
VOCs/$NO_x$ ranging around 4-15 indicates a transitional regime, where ozone is nearly
equally sensitive to both species (Sillman, 1999). As shown in Fig. S8, (a-h), $O_3$ is
mainly formed under the VOC-limited and the transition regimes in CAPAs. As
shown in Figs. S8(i-l) and S8(m-p), both the surface concentrations of VOCs and $NO_x$
are increased when the impacts of ARF are considered. Thus, the contribution of
CHEM in NOAPI is larger than that in NOALL.

When both impacts of API and ARF are considered, the variation pattern of the

difference in hourly process contribution shown in Fig. 7e is similar to that in Fig. 7c,
which indicates that API is the dominant factor to surface-layer $O_3$ reduction.

Figure 8 presents the vertical profiles of simulated daytime $O_3$ concentrations in

three cases (BASE, NOAPI, and NOALL), and the differences in contributions from
each physical/chemical process to hourly $O_3$ variations caused by API, ARF and the
combined effects averaged over CAPAs from Episode1 to Episode4. As shown in Fig.
8a, the $O_3$ concentration is lower in BASE than that in other two scenarios (NOAPI
and NOALL), especially at the lower 12 levels (below 801.8 m), owing to the impacts
of aerosols (API and/or ARF).

The changes in each process contribution caused by API are presented in Fig. 8b. The contribution from CHEM_DIF is -2.1 ppb h$^{-1}$ for the first seven layers (from 25.6 to 318.5 m). Conversely, the contribution from VMIX_DIF shows a positive value under the 318.5 m (between the first layer to the seventh layer) with the mean value of +1.8 ppb h$^{-1}$. The positive variation in VMIX due to API may be associated with the different vertical gradient of O$_3$ between BASE and NOAPI again. The contributions of changed advections (ADVH_DIF and ADVZ_DIF) are relatively small, with mean values of +0.03 and -0.18 ppb h$^{-1}$ below the first seven layers, which may result from small impact of API on wind filed (Fig. S6(a4-d4)). The net difference is a negative value (-0.45 ppb h$^{-1}$); API leads to O$_3$ reduction not only nearly surface but also aloft.

Figure 8c shows the differences in O$_3$ budget due to ARF. When the ARF is considered, the vertical turbulence is weakened and the development of PBL is inhibited, which makes VMIX_DIF negative at the lower seven layers (below the 318.5 m) with a mean value of -0.69 ppb h$^{-1}$, but the variation in CHEM caused by ARF is positive with a mean value of +0.86 ppb h$^{-1}$. The enhanced O$_3$ precursors due to ARF can promote the chemical production of O$_3$ (Tie et al., 2009; Gao et al., 2018). The changes of ADVZ and ADVH (ADVZ_DIF and ADVH_DIF) caused by ARF are associated with the variations in wind filed. When ARF is considered, the horizontal wind speed is decreased (Fig. S9(a)), which makes ADVH_DIF positive at the lower twelve layers with a mean value of +0.30 ppb h$^{-1}$. However, ADVZ_DIF is negative at these layers with a mean value of -0.26 ppb h$^{-1}$ because aerosol radiative effects decrease the transport of O$_3$ from the upper to lower layers (Fig. S9(b)).

In Fig. 8d, the pattern and magnitude of the differences in process contributions between BASE and NOALL are similar to those caused by API, indicating the dominate contributor of API on O$_3$ changes. The impacts of API on O$_3$ both near the surface and aloft are greater than those of ARF.

Figure S10 and S11 detailed show the influencing mechanism of aerosol-radiation interactions on O$_3$ in each episode. Similar variation characteristics can be found among the four episodes as the mean situation discussed above, with the larger impacts of API on O$_3$ both near the surface and aloft than those of ARF, indicating the

role of API is much larger than that of ARF during all the simulated episodes.

**4.5 Discussions**

We presented above the results from our simulations of multi-pollutant air
pollution episodes. In order to make the conclusion be more general, we carried out
simulations for three additional air pollution conditions, i.e., (1) $PM_{2.5}$ pollution alone
(High_PM, with daily mean $PM_{2.5}$ concentration larger than 75 µg m$^{-3}$), (2) $O_3$
pollution alone (High_$O_3$, with the maximum daily 8-h average $O_3$ concentration
larger than 80 ppb), and (3) neither $PM_{2.5}$ nor $O_3$ exceeded air quality standard
(Low_POL, with daily mean $PM_{2.5}$ and the maximum daily 8-h average $O_3$
concentrations smaller than 75 µg m$^{-3}$ and 80 ppb, respectively). For each condition of
air pollution, we examined two episodes.
Figures S12 and S13 show the temporal variations of observed and simulated
$PM_{2.5}$ and $O_3$ concentrations during 7-12 October 2014 (High_PM_Episode1), 7-11
April 2014 (High_PM_Episode2), 15-21 June 2017 (High_$O_3$_Episode1), 12-17 July
2017 (High_$O_3$_Episode2), 13-18 June 2016 (Low_POL_Episode1), and 13-17 July
2016 (Low_POL_Episode2). The temporal variations of observed $PM_{2.5}$ can be well
captured by the model with IOAs of 0.63, 0.82, 0.56, 0.42, 0.76 and 0.54, and NMBs
of 7.4%, 20.3%, -21.7%, -25.9%, 14.7% and -29.3% during High_PM_Episode1,
High_PM_Episode2, High_$O_3$_Episode1, High_$O_3$_Episode2, Low_POL_Episode1,
and Low_POL_Episode2, respectively. The model also simulates well the diurnal
variation of $O_3$ over the North China, with IOAs of 0.87, 0.80, 0.87, 0.90, 0.84 and
0.86, and NMBs of -9.4%, -29.5%, -15.2%, -9.4%, 11.6% and 18.0% in these six
episodes, respectively.
Figure 9 shows changes in daytime surface-layer $O_3$ due to API, ARF, and the
combined effects (denoted as ALL) of High_PM_Episode1, High_PM_Episode2,
High_$O_3$_Episode1,         High_$O_3$_Episode2,         Low_POL_Episode1,         and
Low_POL_Episode2. As summarized in Table 2, all the simulations confirm the same
conclusion that the reduction in $O_3$ by API is larger than that by ARF. Averaged over
the entire domain, the percentage reductions in $O_3$ by API and ARF are, respectively,
29.3% and 6.2% in High_PM_Episode1, 16.9% and 4.7% in High_PM_Episode2, 5.3%
and 0.1% in High_O$_3$_Episode1, 4.5% and 0.1% in High_O$_3$_Episode2, 6.8% and 1.0%
in Low_POL_Episode1, and 2.9% and 0.7% in Low_POL_Episode2. It's worth
noting that the percentage reductions in O$_3$ from both API and ARF in High_PM
episodes are 1.6~3.2 times the impacts in the complex episodes, while the impacts in
cases of Low_POL and High_O$_3$ are 0.3~0.7 times the impacts of complex episodes."

## 5 Conclusions

In this study, the fully coupled regional chemistry transport model WRF-Chem is
applied to investigate the impacts of aerosol-radiation interactions, including the
impacts of aerosol-photolysis interaction (API) and the impacts of aerosol-radiation
feedback (ARF), on O$_3$ during summertime complex air pollution episodes during 28
July to 3 August 2014 (Episode1), 8-13 July 2015 (Episode2), 5-11 June 2016
(Episode3) and 28 June to 3 July 2017 (Episode4). Three sensitivity experiments are
designed to quantify the respective and combined impacts from API and ARF.
Generally, the spatiotemporal distributions of observed pollutant concentrations and
meteorological parameters can be captured fairly well by the model with index of
agreement of 0.44~0.94 for pollutant concentrations and 0.70~0.99 for meteorological
parameters.
Sensitivity experiments show that aerosol-radiation interactions decrease
BOT_SW, WS$_{10}$, PBLH, J[NO$_2$], and J[O$^1$D] by 92.4~102.9 W m$^{-2}$, 0.05~0.15 m s$^{-1}$,
129.0~249.0 m, $1.8 \times 10^{-3}$~$2.0 \times 10^{-3}$ s$^{-1}$, and $5.7 \times 10^{-6}$~$6.4 \times 10^{-6}$ s$^{-1}$ over CAPAs,
and increase ATM_SW by 72.8~85.2 W m$^{-2}$, respectively. The changed
meteorological variables and weakened photochemistry reaction further reduce
surface-layer O$_3$ concentrations by 10.0~11.4 ppb, with relative changes of
74.6%~106.5% by API and of -6.5%~25.4% by ARF.
We further examine the influencing mechanism of aerosol-radiation interactions
on O$_3$ by using integrated process rate analysis. API can directly affect O$_3$ by reducing
the photochemistry reactions within the lower several hundred meters and therefore
amplify the O$_3$ vertical gradient, which promotes the vertical mixing of O$_3$. The
reduced photochemistry reactions of $O_3$ weaken the chemical contribution and reduce
surface $O_3$ concentrations, even though the enhanced vertical mixing can partly
counteract the reduction. ARF affects $O_3$ concentrations indirectly through the
changed meteorological variables, e.g., the decreased PBLH. The suppressed PBL can
weaken the vertical mixing of $O_3$ by turbulence. Generally, the impacts of API on $O_3$
both near the surface and aloft are greater than those of ARF, indicating the dominant
role of API on $O_3$ reduction related with aerosol-radiation interactions.

This study provides a detailed understanding of aerosol impacts on $O_3$ through

aerosol-radiation interactions (including both API and ARF), with the general
conclusion summarized as follows: when the impacts of aerosol-radiation interactions
are considered, the changed meteorological variables and weakened photochemistry
reaction can change surface-layer $O_3$ concentrations during the warm season, and the
API is the dominant factor for $O_3$ reduction. The results can also imply that future
$PM_{2.5}$ reductions may lead to $O_3$ increases due to weakened aerosol-radiation
interactions. A recent study emphasized the need for controlling VOCs emissions to
mitigate $O_3$ pollution (Li et al., 2019b). Therefore, tighter controls of $O_3$ precursors
(especially VOCs emissions) are needed to counteract future $O_3$ increases caused by
weakened aerosol-radiation interactions.

There are some limitations in this work. (1) In the current CBMZ and MOSAIC

schemes, the formation of SOA (secondary organic aerosol) is not included (Gao et al.,
2015; Chen et al., 2019). The absence of SOA can underestimate the impacts of API
and ARF on $O_3$. Meanwhile, the lack of SOA may lead to weaker heterogeneous
reactions to result in higher $O_3$ concentrations (Li et al., 2019c). The net effect of the
two processes will be discussed and quantified in our future study. (2) The CNEMC
network was built in 2013. Before 2013, the national observations of $PM_{2.5}$ and $O_3$
concentrations were not available, which make it difficult to select the time and
location of complex air pollution events and to evaluate the model results. Based on
observation data, we were mainly focused on impacts of ARF and API on surface $O_3$
for complex air pollution episodes from 2014 to 2017. Additional simulations of
High_PM, High_$O_3$, and Low_POL support the conclusion obtained from the
complex air pollution episodes that the reduction in $O_3$ by API is larger than that by
ARF.

## Data availability

The observed hourly surface concentrations of air pollutants are derived from the China National Environmental Monitoring Center (http://www.cnemc.cn). The observed surface meteorological data are obtained from NOAA's National Climatic Data Center (https://gis.ncdc.noaa.gov/maps/ncei/cdo/hourly). The radiosonde data are provided by the University of Wyoming (http://weather.uwyo.edu/). The photolysis rates of nitrogen dioxide in Beijing are provided by Xin Li (li_xin@pku.edu.cn). The aerosol optical depth in Beijing is obtained from the AERONET level 2.0 data collection (http://aeronet.gsfc.nasa.gov/). The simulation results can be accessed by contacting Lei Chen (chenlei@nuist.edu.cn) and Hong Liao (hongliao@nuist.edu.cn).

## Author contributions

HY, LC, and HL conceived the study and designed the experiments. HY and LC performed the simulations and carried out the data analysis. JZ, WW, and XL provided useful comments on the paper. HY, LC, and HL prepared the paper with contributions from all co-authors.

## Competing interests

The authors declare that they have no competing interests.

## Acknowledgements

We acknowledge the computing resources from the University-Industry Collaborative Education Program between the Ministry of Education and Huawei.

## Financial support

This research has been supported by National Natural Science Foundation of China (grant nos. 42021004, 91744311, 42007195), the Meteorological Soft Science

Program of China Meteorological Administration (2021ZZXM46), and the
Postgraduate Research and Practice Innovation Program of Jiangsu Province
(KYCX21_1014).

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

**Table 1.** Physical parameterization options used in the simulation.

| Options | Schemes |
| --- | --- |
| Microphysics scheme | Lin (Purdue) scheme (Lin et al.,1983) |
| Cumulus scheme | Grell 3D ensemble scheme |
| Boundary layer scheme | Yonsei University PBL scheme (Hong et al., 2006) |
| Surface layer scheme | Monin-Obukhov surface scheme (Foken, 2006) |
| Land-surface scheme | Unified Noah land-surface model (Chen and Dudhia, 2001) |
| Longwave radiation scheme | RRTMG (Iacono et al., 2008) |
| Shortwave radiation scheme | RRTMG (Iacono et al., 2008) |

**Table 2.** Detailed information of the analyzed episodes, including the impacts of API, ARF and ALL on $O_3$ concentrations under different air pollution conditions. The numbers in bold indicate the concentrations exceeded the Class II limit of the National Ambient Air Quality Standards of China. The numbers in parentheses indicate the percentage changes in $O_3$ concentration.

| Type | Episode | Time | PM$_{2.5}$ pollution ($\mu g\ m^{-3}$) | O$_3$ pollution (ppb) | API (ppb) | ARF (ppb) | ALL (ppb) |
|---|---|---|---|---|---|---|---|
| **Complex air pollution** | Episode1 | 2014.7.28-2014.8.3 | **113.3** | **80.0** | -8.5 (-10.2%) | -2.9 (-3.2%) | -11.4 (-13.7%) |
| | Episode2 | 2015.7.8-2015.7.13 | **79.3** | **89.6** | -10.3 (-11.8%) | -1.0 (-1.1%) | -11.3 (-13.0%) |
| | Episode3 | 2016.6.5-2016.6.11 | **76.5** | **87.6** | -9.1 (-11.2%) | -0.9 (-1.0%) | -10.0 (-12.3%) |
| | Episode4 | 2017.6.28-2017.7.3 | **75.4** | **113.8** | -11.4 (-12.2%) | 0.7 (0.5%) | -10.7 (-11.6%) |
| **High_PM** | Episode1 | 2014.10.7-2014.10.12 | **223.5** | 46.9 | -15.3 (-29.3%) | -3.9 (-6.2%) | -19.2 (-37.6%) |
| | Episode2 | 2014.4.7-2014.4.11 | **111.7** | 54.8 | -7.3 (-16.9%) | -2.4 (-4.7%) | -9.7 (-22.6%) |
| **High_O$_3$** | Episode1 | 2017.6.15-2017.6.21 | 61.9 | **103.6** | -4.5 (-5.3%) | -0.1 (-0.1%) | -4.6 (-5.5%) |
| | Episode2 | 2017.7.12-2017.7.17 | 45.6 | **100.4** | -3.8 (-4.5%) | -0.1 (-0.1%) | -3.9 (-4.6%) |
| **Low_POL** | Episode1 | 2016.6.13-2016.6.18 | 36.5 | 62.4 | -4.4 (-6.8%) | -0.6 (-1.0%) | -5.0 (-7.9%) |
| | Episode2 | 2016.7.13-2016.7.17 | 38.3 | 55.9 | -1.9 (-2.9%) | -0.5 (-0.7%) | -2.4 (-3.7%) |

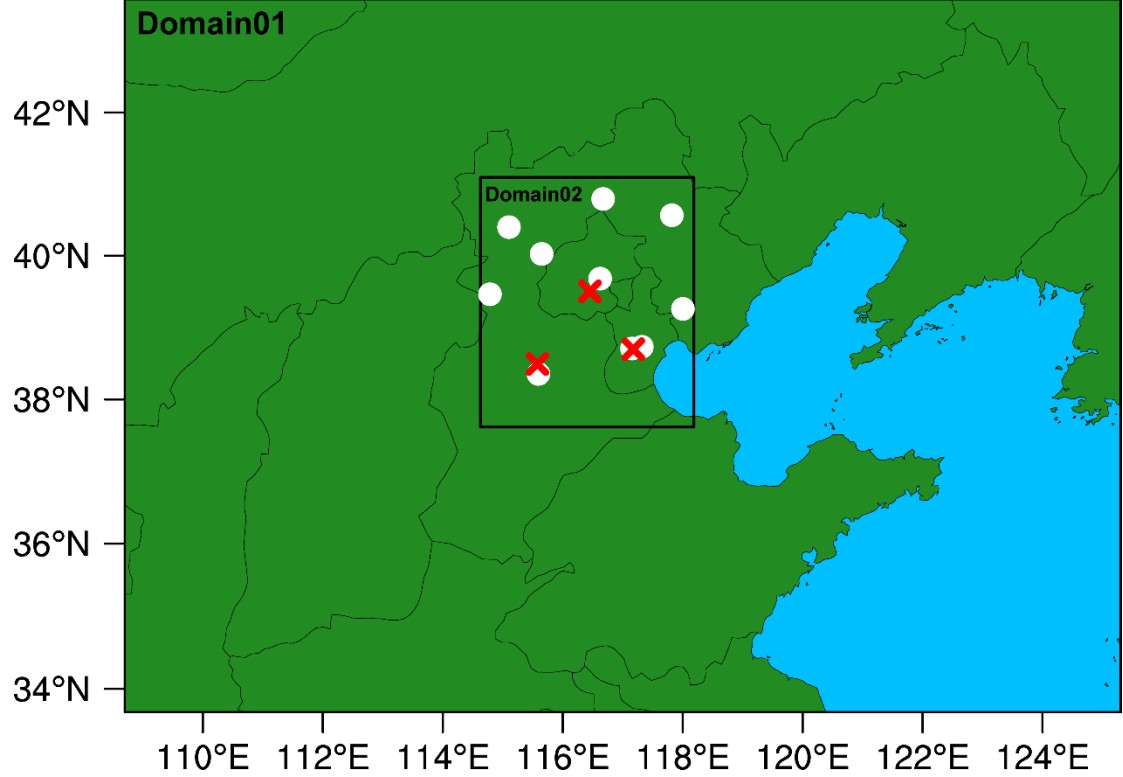

**Figure 1.** Map of the two WRF-Chem modeling domains with the locations of
meteorological (white dots) and environmental (red crosses) observation sites used for
model evaluation.

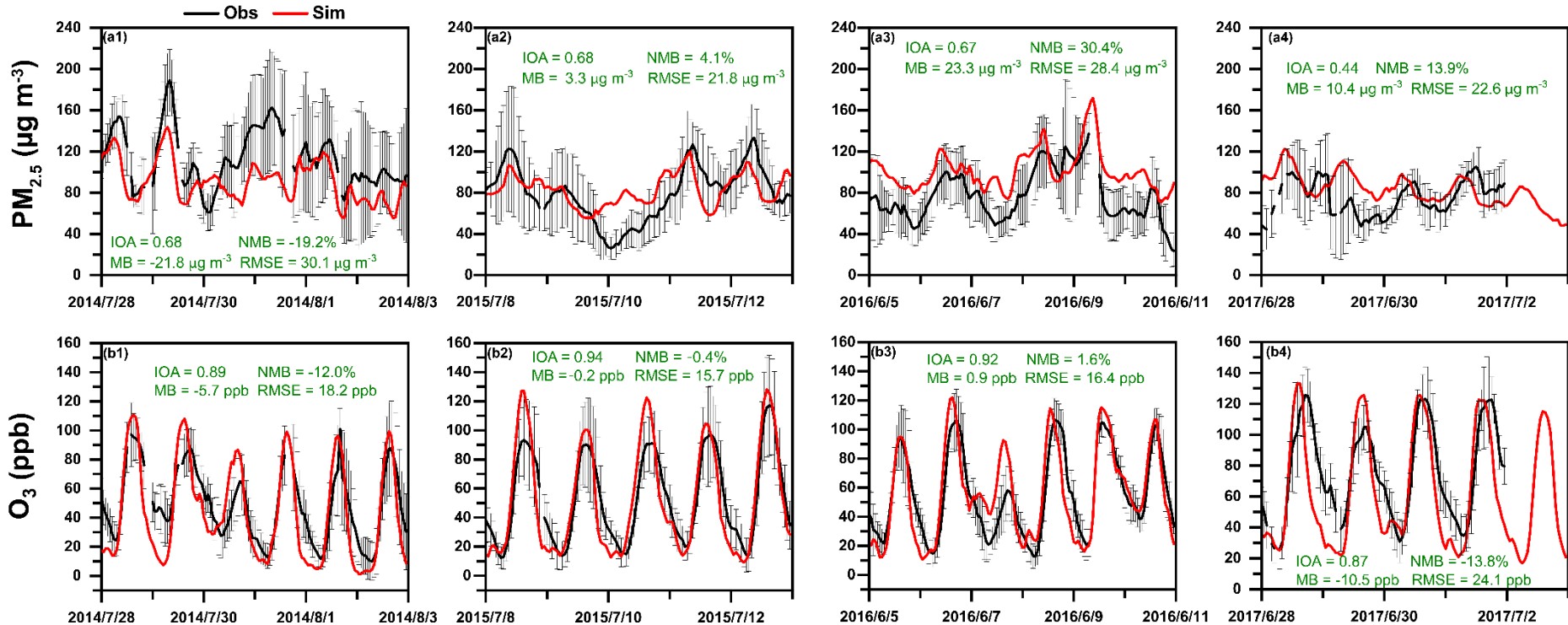

**Figure 2.** Time series of observed (black) and simulated (red) hourly surface (a) $PM_{2.5}$ and (b) $O_3$ concentrations averaged over the thirty-two observation sites in Beijing, Tianjin, and Baoding during 28 July to 3 August 2014 (Episode1, a1-b1), 8-13 July 2015 (Episode2, a2-b2), 5-11 June 2016 (Episode3, a3-b3) and 28 June to 3 July 2017 (Episode4, a4-b4). The error bars represent the standard deviations. The calculated index of agreement (IOA), mean bias (MB), normalized mean bias (NMB) and root-mean-square error (RMSE) are also shown.

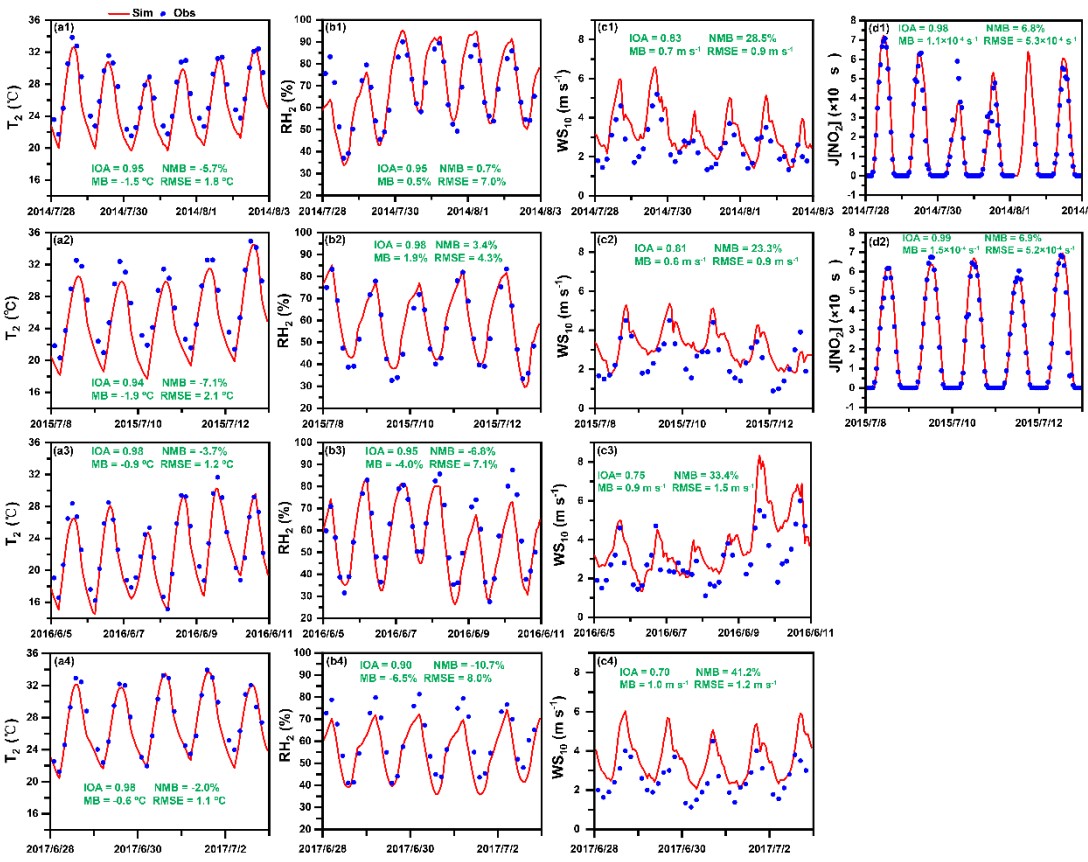

Figure 3. Time series of 3-hourly observed (blue dots) and hourly simulated (red lines) (a) 2-m temperature ($T_2$), (b) 2-m relative humidity ($RH_2$), (c) wind speed at 10 m ($WS_{10}$) averaged over ten meteorological observation stations, and (d) surface photolysis rate of $NO_2$ ($J[NO_2]$) during 28 July to 3 August 2014 (Episode1, a1-d1), 8-13 July 2015 (Episode2, a2-d2), 5-11 June 2016 (Episode3, a3-c3) and 28 June to 3 July 2017 (Episode4, a4-c4). The calculated index of agreement (IOA), mean bias (MB), normalized mean bias (NMB) and root-mean-square error (RMSE) are also shown.

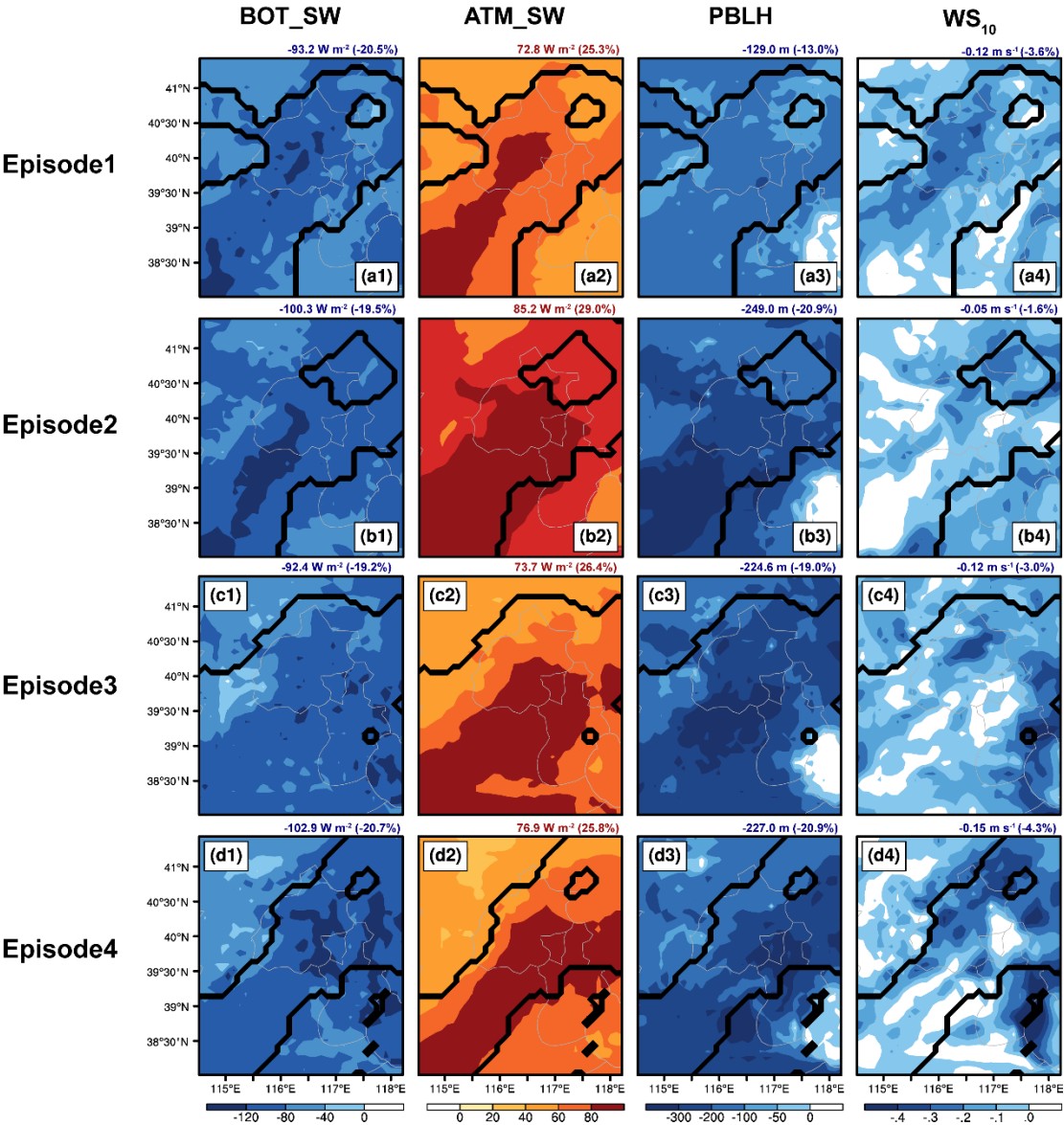

**Figure 4.** The impacts of aerosol-radiation interactions on shortwave radiation at the surface (BOT_SW), shortwave radiation in the atmosphere (ATM_SW), PBL height (PBLH), and 10-m wind speed (WS$_{10}$) in the daytime (08:00-17:00 LST) during 28 July to 3 August 2014 (Episode1), 8-13 July 2015 (Episode2), 5-11 June 2016 (Episode3) and 28 June to 3 July 2017 (Episode4). The regions sandwiched between two black lines are defined as the complex air pollution areas (CAPAs) where the mean daily PM$_{2.5}$ and MDA8 O$_3$ concentrations in BASE case are larger than 75 µg m$^{-3}$ and 80 ppb. The calculated changes (percentage changes) averaged over CAPAs are also shown at the top of each panel.

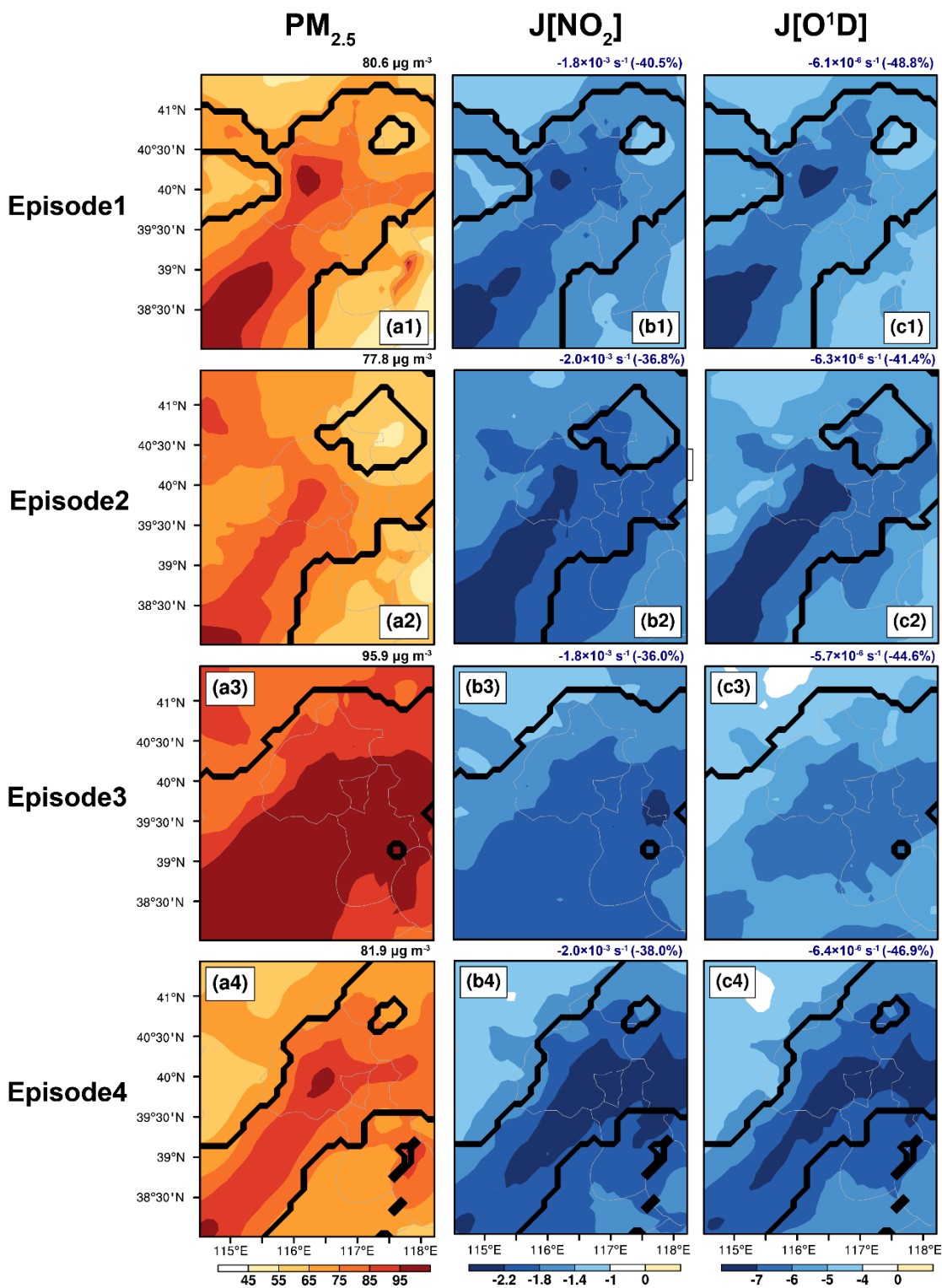

**Figure 5.** Spatial distributions of (a) simulated surface-layer $PM_{2.5}$ concentrations in BASE cases, and the changes in surface (b) $J[NO_2]$ and (c) $J[O^1D]$ due to aerosol-radiation interactions in the daytime (08:00-17:00 LST) during 28 July to 3 August 2014 (Episode1), 8-13 July 2015 (Episode2), 5-11 June 2016 (Episode3) and 28 June to 3 July 2017 (Episode4). The calculated values (percentage changes) avaraged over CAPAs are also shown at the top of each panel.

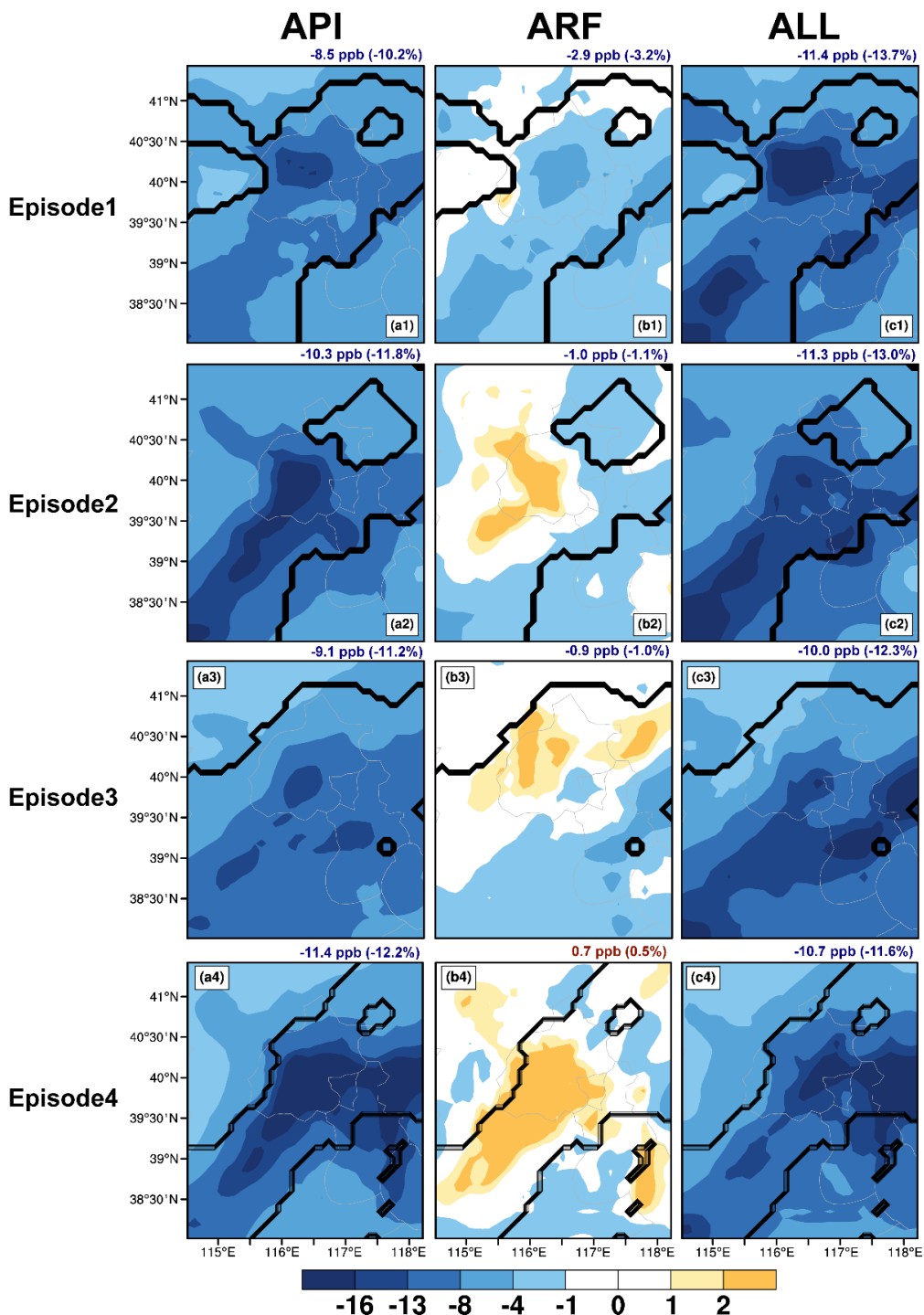

**Figure 6.** The changes in surface-layer ozone due to (a) aerosol-photolysis interaction (API), (b) aerosol-radiation feedback (ARF), and (c) the combined effects (ALL, defined as API+ARF) in the daytime (08:00-17:00 LST) during 28 July to 3 August 2014 (Episode1), 8-13 July 2015 (Episode2), 5-11 June 2016 (Episode3) and 28 June to 3 July 2017 (Episode4). The changes (percentage changes) in $O_3$ concentrations caused by API, ARF and ALL avaraged over CAPAs are also shown at the top of each panel.

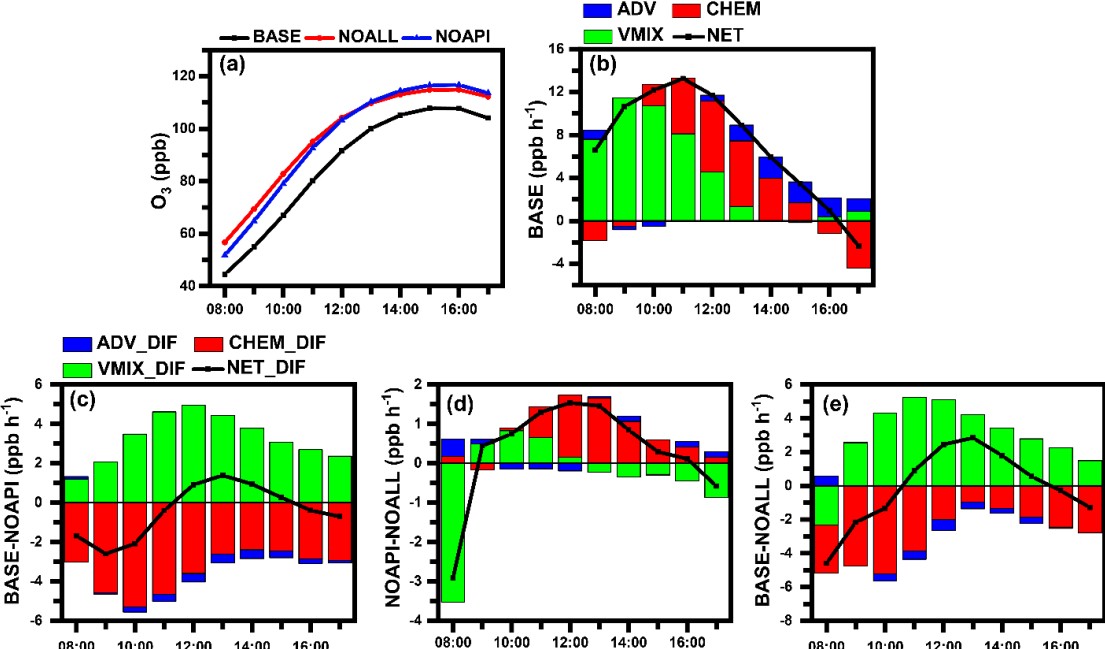

**Figure 7.** Temporal evolution characteristics of aerosol-radiation interactions on $O_3$ averaged over the four episodes. (a) Diurnal variations of simulated surface $O_3$ concentrations in BASE (black dotted line), NOAPI (blue dotted line), and NOALL (red dotted line) cases over CAPAs. (b) The hourly surface $O_3$ changes induced by each physical/chemical process using the IPR analysis method in BASE case. (c-e) Changes in hourly surface $O_3$ process contributions caused by API (BASE minus NOAPI), ARF (NOAPI minus NOALL), and ALL (BASE minus NOALL) over CAPAs during the daytime (08:00-17:00 LST). The black lines with squares denote the net contribution of all processes (NET, defined as VMIX+CHEM+ADV). Differences of each process contribution are denoted as VMIX_DIF, CHEM_DIF, ADV_DIF, and NET_DIF.

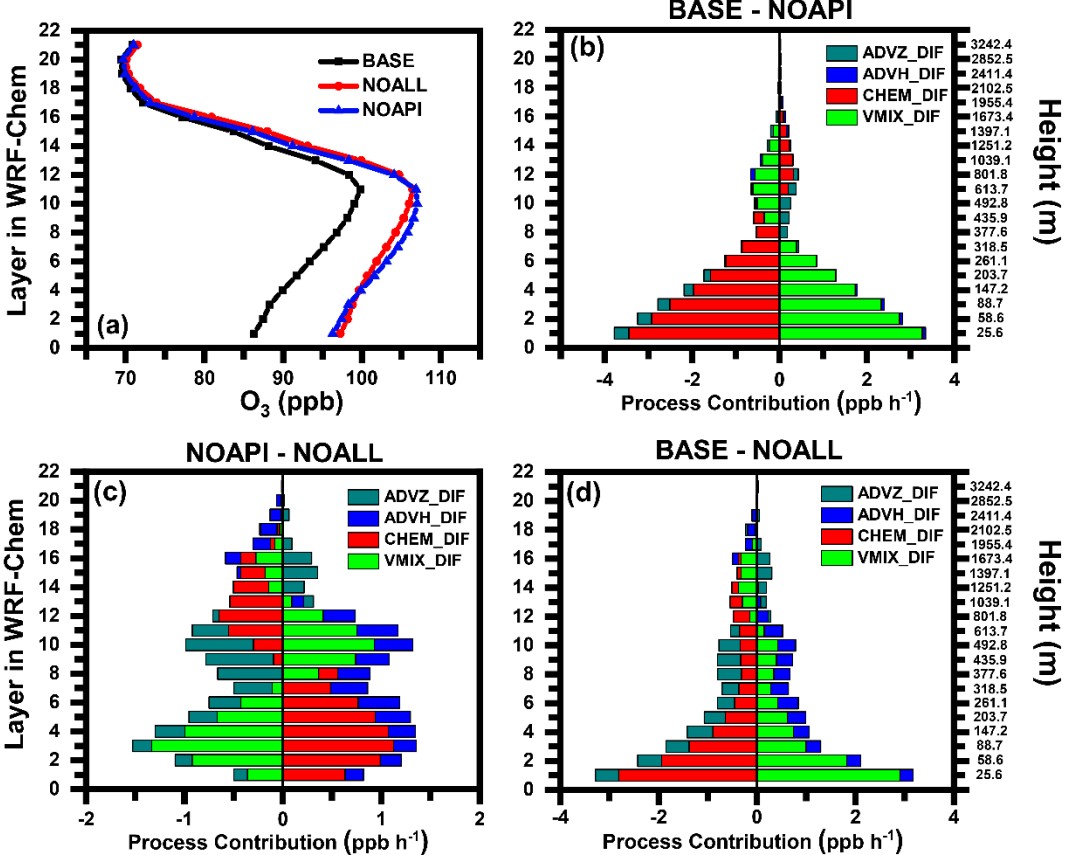

**Figure 8.** The impacts of aerosol-radiation interactions on vertical $O_3$ averaged over
the four episodes. (a) Vertical profiles of simulated $O_3$ concentrations in BASE (black
dotted line), NOAPI (blue dotted line), and NOALL (red dotted line) cases over
CAPAs. (b-d) Changes in $O_3$ budget due to API, ARF, and ALL over CAPAs during
the daytime (08:00-17:00 LST). Differences of each process contribution are denoted
by ADVZ_DIF, ADVH_DIF, CHEM_DIF, and VMIX_DIF.

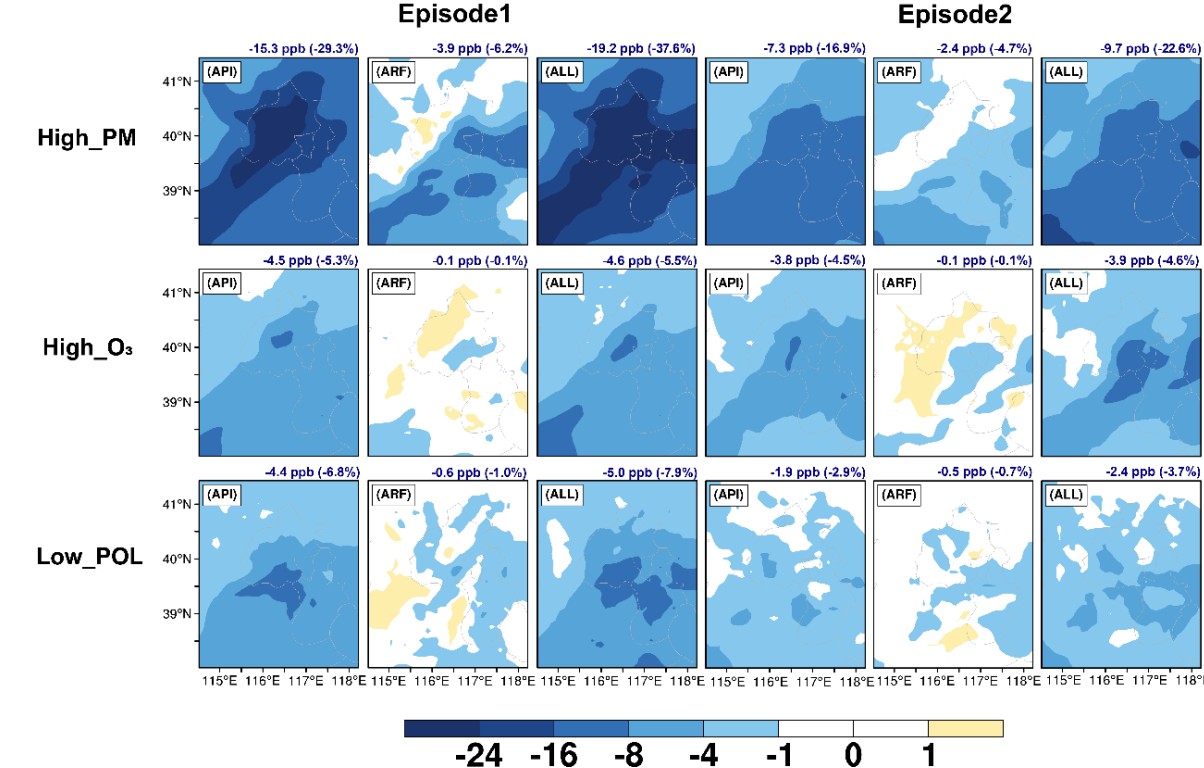

**Figure 9.** The changes in surface-layer $O_3$ due to aerosol-photolysis interaction (API), aerosol-radiation feedback (ARF), and the combined effects (ALL, API+ARF) in the daytime (08:00-17:00 LST) of 7-12 October 2014 (High_PM_Episode1), 7-11 April 2014 (High_PM_Episode2), 13-18 June 2016 (Low_POL_Episode1), 13-17 July 2016 (Low_POL_Episode2), 15-21 June 2017 (High_O_3_Episode1), and 12-17 July 2017 (High_O_3_Episode2). The changes (percentage changes) in $O_3$ concentrations caused by API, ARF and ALL avaraged over the entire simulated domain are also shown at the top of each panel.