# Peer review of "Impacts of aerosol-photolysis interaction and aerosol-radiation"

_Atmospheric Chemistry and Physics, 2021_

## Referee Comment (RC2)

**Impacts of aerosol-photolysis interaction and aerosol-radiation feedback on surface-layer ozone in North China during a multi-pollutant air pollution episode**

In this study, Yang et al. investigate the impact of aerosol-radiation interactions on $O_3$ formation during a multi-pollutant air pollution episode in Northern China. Additionally, the study uses process analysis to analyze how the aerosol-radiation interactions affect $O_3$ through various physical and chemical mechanisms. This is an interesting research topic with valid research methods and an overall well written and well-structured manuscript. However, the period of analysis is far too short (i.e., 7 days) to robustly quantify the impact of aerosol-radiation impacts in this region or to describe any variability. Additionally, the time period analyzed appears somewhat arbitrary and is nearly a decade removed from current conditions. For these reasons, the manuscript is not currently at the scientific level of the Atmospheric Chemistry and Physics Journal. However, this manuscript would be suitable for publication in ACP if either it is restructured to focus on how the methods used are unique and different from past work or if the authors investigate longer periods to generate more robust analysis and conclusions. Please find my major and minor comments below.

Major Comments:

1) The novelty of this study is that it is the first time that API and ARF are investigated for synchronous occurrences of high $PM_{2.5}$ and $O_3$ concentrations. This is a rather broad research question to be focused on only one region and one very minor time period. Why do the authors not conduct simulations for either several of these small pollution episodes in this region or for similar episodes in other locations in China?

2) Given that government controls have substantially changed emissions in the last decade and will continue into future, how will this research remain relevant in the future or how relevant is it to today's air pollution in China, since the period examined is 7 years ago?

3) Is the focus of this research the method in which API and ARF are investigated or the impact of API and ARF in North China? If it is the former than the authors need to reword the abstract, conclusions, and objectives to make it clear that this study is a "proof-of-concept" study on how to best investigate API and ARF in high $O_3$ and $PM_{2.5}$ episodes. If the focus is the latter, the authors need to do additional simulations of other high multi-pollutant episodes, perhaps some closer to current conditions and others in the mid 2000s to see if there is change over time or to make the analysis and conclusions more robust.

4) Does this version of WRF-Chem's CBM-Z and MOSAIC modules have a volatility basis set (VBS) option to simulate secondary organic aerosols and if so is it used? Given that, this is a high $O_3$ and $PM_{2.5}$ episode there should be a substantial amount of secondary organic aerosol from abundant oxidants and precursors that may be missed in the model

without an advanced SOA scheme. How do the author's address the impact of SOA on their conclusions?

5) The authors are investigating aerosol radiation interactions, but the authors do not evaluate the model's performance against either radiation balance datasets or aerosol optical depth. Since these parameters are more important than surface evaluations of air pollutants to understanding API and ARF, the authors should evaluate their model configuration against satellite AOD and radiation variables such as MODIS or CERES-EBAF.

6) Are there only three meteorological observation stations in the domain against? If so, why do the authors not also validate their meteorological performance against gridded products like the Climate Research Unit (CRU) datasets to ensure their performance statistics are robust?

7) Given that interactions between $O_3$ and $PM_{2.5}$ are non-linear, how do the authors justify using a simple ratio value (i.e., ROP) to relate these interactions? If this ratio does not account for non-linearity, how useful is this value?

8) The axis labels and legends of Figure 7 are difficult to read. Either each panel should be larger overall or the font sizes of the axes and legends need increased.

Minor Comments:

1) In the abstract, there is no context for the values listed. Further reading into the manuscript reveals that these values are the averages in the areas of the complex air pollution areas. The authors should briefly state that these values are for daytime average changes in complex air pollution areas in the abstract.

I would also suggest adding a more processed based explanation of the changes in atmospheric state rather than simply listing a long series of values. For example, the authors could state something similar to the following:

"Aerosol radiation interactions lead to shortwave dimming at the earth's surface of X, which reduce photolysis rates by X. The dimming stabilizes the atmosphere via surface cooling of X, which reduces PBL height by X. The stabilized atmosphere increases saturation in the lower atmosphere by X. etc…."

2) Make it clear throughout the manuscript when you are referring to surface level $O_3$ and $PM_{2.5}$.

3) Lines 179-181: The missing $PM_{2.5}$ could also be from missing SOA formation pathways, as mentioned above, if no advanced SOA formulations are used.

4) Is "downward shortwave radiation in the atmosphere" the SWDNT variable from WRF-Chem? If so, the name of this variable is "downward shortwave radiation at the top of the atmosphere".

5) Lines 217-218: If ATM_SW is the SWDNT variable, what is causing it to increase? SWDNT is usually controlled by the solar constant. Is it possible this is reflected upward shortwave (SWUPT)?

6) Lines 248-249: This should be revised to make it clearer that ARF primarily impacts $O_3$ through changing the NOx distribution.

7) Lines 270-281: Is VMIX increasing surface $O_3$ because it is mixing down higher $O_3$ concentrations from aloft or because vertical mixing is suppressed due to a stable atmosphere?

8) Lines 282-294: Why does the VMIX contribution increase because of API?

9) Lines 295-301: Explain why VMIX_DIF and CHEM_DIF are positive during the day due to ARF.

10) Lines 315-316: Explain how different vertical $O_3$ gradients can cause this change.

Line Comments:

1) Line 49: This should be "Earth's radiative balance" or "Earth's energy balance"

2) Lines 54-56: Are these studies all focused on China? If so, state that in the sentence. Change "were" to "are".

3) Lines 56-63: State the domain and time period of Gao et al., (2015) at the beginning of this statement rather than the end

4) Line 66: Add "the" before North China Plain

5) Lines 66-67: If this is referring to surface $PM_{2.5}$ concentrations, add "surface" before $PM_{2.5}$ concentrations.

6) Line 204: should be "attention"

7)  Line 256: Center align the equation.

8) Line: 259: Why are there parentheses in the units?

9)  Lines 288-289: This sentence is a little confusing. Is Net_DIF the sum of CHEM_DIF, VMIX_DIF, and ADV_DIF? If so, state that explicitly and then indicate what Net_DIF describes.

10) Line 321: Remove "in the"

11) Line 361: Remove "the contribution from VMIX and"

12) Line 373: Either "A recent study" or "Recent studies have"

---

## Author Comment (AC1)

**Response to Comments of Reviewer #1**

**(comments in *italics*)**

**Manuscript number:** acp-2021-119

**Title:** Impacts of aerosol-photolysis interaction and aerosol-radiation feedback on surface-layer ozone in North China during a multi-pollutant air pollution episode

*Yang et al. examined the impacts of aerosols on surface ozone through the two well-known pathways, i.e., aerosol-photolysis interaction and aerosol-radiation feedback. The novelty of this study is its focus on the polluted episodes with elevated both PM$_{2.5}$ and ozone levels over North China. They also quantified the chemical and physical processes that drive the aerosol-radiation interactions.*

*Overall, this is a timely study and it clearly demonstrates the impacts of aerosols on ozone pollution. The structure of this manuscript is easy to follow. Although some of the manuscript needs further clarification, the results are generally convincing. As such, I think it is publishable after the following issues are addressed.*

**Response:**

Thanks to the reviewer for the valuable comments and suggestions which are very helpful for us to improve our manuscript. We have revised the manuscript carefully, as described in our point-to-point responses to the comments.

**Specific Comments:**

*1. In Abstract: ozone changes refer to MDA8 ozone or daytime ozone?*

**Response:**

The ozone changes in abstract is daytime ozone. According to the reviewer's comments, we have added this information in the revised manuscript. (**Page 2, Line 33**)

*2. Line 177: a correlation coefficient of 0.66 reads like not high!*

**Response:**

According to the reviewer's suggestion, we have corrected it in the revised manuscript as follows: "The model can also reasonably capture the temporal variations of observed PM$_{2.5}$ and O$_3$ with correlation coefficients (R) of 0.66 for PM$_{2.5}$ and 0.86 for O$_3$." (**Page 7, Line 180-183**)

*3. Lines 179-181: the oxidation of SO$_2$ by NO$_2$ in aqueous aerosols is important for summertime?*

**Response:**

Thanks for your suggestion, we have changed the explanation in the revised manuscript as follows: "The failure to reproduce PM$_{2.5}$ peak values may be attributed to incomplete treatments of chemical reactions in WRF-Chem, e.g., missing the heterogeneous chemistry in the model (Cheng et al., 2016) and the lack of secondary organic aerosols (SOA) formation pathways in the aerosol module (Chen et al., 2019)." (**Page 7-8, Line 183-188**)

4. *Lines 248-251: this statement looks reasonable here, but in the later text the process analysis shows that chemistry will be enhanced by ARF. Instead, ARF decreases ozone through physical processes.*

**Response:**

Thanks for the reviewer's suggestion. We have deleted this sentence in the revised manuscript.

5. *Line 260: "is" should be "are". Please do proof-reading throughout the text.*

**Response:**

This sentence has been deleted in the revised manuscript. According to the reviewer's comments, proof-reading has been conducted through the whole revised manuscript.

6. *Line 310: It is Okay to use model levels (e.g., 12 levels), but it will be better to add model height in meters as well.*

**Response:**

Thanks for reviewer's suggestion. We have added the model height in meters in the revised manuscript. (**Page 12, Line 314, Line 317-319, Line 328**)

7. *Lines 326-327: why do you need this statement?*

**Response:**

Analyzing Fig. 8c we can conclude that ARF promotes the $O_3$ chemical production with a positive mean value of 0.66 ppb $h^{-1}$. The enhanced $O_3$ precursors due to ARF can promote the chemical production of $O_3$. According to the reviewer's comment, we have deleted this statement in the revised manuscript.

8. *Lines 327-328: Please provide evidence to support this conclusion.*

**Response:**

The typical VOCs/$NO_x$ ratio is calculated to classify sensitivity regimes and to indicate the possible $O_3$ responses to changes in VOCs and/or $NO_x$ concentrations. $O_3$ production is VOC-limited if the ratio is less than 4, and it is $NO_x$-limited if the ratio is larger than 15 (Edson et al., 2017; Li et al., 2017). The ratio of VOCs/$NO_x$ ranging around 4-15 indicates a transitional regime, where ozone is nearly equally sensitive to each species (Sillman, 1999). As shown in Fig. R1(a-c), $O_3$ are mainly formed under VOC-limited and transition regimes in CAPAs, which means that the increased concentrations of VOCs and $NO_x$ are favorable for ozone chemical production. As shown in Fig. R1(e) and (h), both the surface concentrations of VOCs and $NO_x$ are increased when the impacts of ARF are considered. Thus, the contribution of CHEM in NOAPI is larger than that in NOALL. Similar results can also be found in Gao et al. (2018).

[Figure]

**Figure R1.** The ratios of VOCs/NO$_x$ calculated from (a) BASE, (b) NOALL, and (c) NOAPI. The changed surface-layer concentrations of VOCs and NO$_x$ (NO$_2$+NO, ppb) caused by (d, g) API, (e, h) ARF, and (f, i) ALL during the daytime (08:00-17:00 LST) from 28 July to 3 August 2014. The calculated values averaged over CAPAs are also shown at the top of each panel.

9. *Discussion. I think the authors should do some comparisons between your results with previous studies. This is important for readers to better understand your case study results. Moreover, how about the applicability of the calculated ROP of -0.14 ppb (µg m$^{-3}$)$^{-1}$?*

**Response:**

According to the comments of Reviewer#2, another two complex air pollution episodes (8-13 July 2015 and 5-11 June 2016) in this region are selected to conduct simulations for generating general conclusions (**Page 13-14, Line 343-365**). Meanwhile, a discussion about the impacts of secondary organic aerosols (SOA) is also added in the section 6 (**Page 15, Line 402-412**).

Thanks to the reviewer's comments. As the relationship between O$_3$ and PM$_{2.5}$ is non-linear, and the simple index of ROP can not fully represent the impacts of aerosols on surface O$_3$, so we

delete the ROP in the revised manuscript.

*10. Fig.2: It will be better to add error bars for observed PM$_{2.5}$ and ozone.*
**Response:**

    According to the reviewer's suggestion, error bars have been added in Fig. 2 in the revised manuscript. (**Page 27**)

*11. Fig.3: what are the cities these plots for?*
**Response:**

    The averaged T$_2$, RH$_2$, and WS$_{10}$ are collected from ten meteorological observation stations, and the detail information about the sites is listed in Table S1. The photolysis rates of NO$_2$ (J[NO$_2$]) are observed in Peking University. More details are explained in section 2.3. (**Page 6**)

*12. Fig.7: what are the layers your process analysis applied for? I don't see this key information here, as well as in the text.*
**Response:**

    The surface-layer, namely, first-layer O$_3$ concentrations are analyzed in Fig. 7. Thanks for reviewer's suggestion, we have added this information in the revised manuscript. (**Page 10, Line 266**)

**Reference:**

Chen, L., Zhu, J., Liao, H., Gao, Y., Qiu, Y., Zhang, M., Liu, Z., Li, N., and Wang, Y.: Assessing the formation and evolution mechanisms of severe haze pollution in the Beijing–Tianjin–Hebei region using process analysis, Atmos. Chem. Phys., 19, 10845–10864, https://doi.org/10.5194/acp-19-10845-2019, 2019.

Cheng, Y., Zheng, G., Chao, W., Mu, Q., Bo, Z., Wang, Z., Meng, G., Qiang, Z., He, K., and Carmichael, G.: Reactive nitrogen chemistry in aerosol water as a source of sulfate during haze events in China, Science Advances, 2, https://doi.org/10.1126/sciadv.1601530, 2016.

Edson, C. T., Ivan, H.-P. and Alberto, M.: Use of combined observational- and model-derived photochemical indicators to assess the O$_3$-NO$_x$-VOC System sensitivity in urban areas, Atmosphere., 8, 22. https://doi.org/10.3390/ atmos8020022, 2017.

Li, K., Chen, L., Ying, F., White, S. J., Jang, C., Wu, X., Gao, X., Hong, S., Shen, J., Azzi, M. and Cen, K: Meteorological and chemical impacts on ozone formation: a case study in Hangzhou, China, Atmos. Res., 196, https://doi.org/10.1016/ j.atmosres.2017.06.003, 2017.

Sillman, S.: The relation between ozone, NO$_x$ and hydrocarbons in urban and polluted rural environments, Atmos. Environ., 33, 1821-1845, https://doi.org/ 10.1016/S1352-2310(98)00345-8, 1999.

**Thank you very much for your comments and suggestions.**

---

## Author Comment (AC2)

**Response to Comments of Reviewer #2**

**(comments in *italics*)**

**Manuscript number:** acp-2021-119

**Title:** Impacts of aerosol-photolysis interaction and aerosol-radiation feedback on surface-layer ozone in North China during a multi-pollutant air pollution episode

*In this study, Yang et al. investigate the impact of aerosol-radiation interactions on $O_3$ formation during a multi-pollutant air pollution episode in Northern China. Additionally, the study uses process analysis to analyze how the aerosol-radiation interactions affect $O_3$ through various physical and chemical mechanisms. This is an interesting research topic with valid research methods and an overall well written and well-structured manuscript. However, the period of analysis is far too short (i.e., 7 days) to robustly quantify the impact of aerosol-radiation impacts in this region or to describe any variability. Additionally, the time period analyzed appears somewhat arbitrary and is nearly a decade removed from current conditions. For these reasons, the manuscript is not currently at the scientific level of the Atmospheric Chemistry and Physics Journal. However, this manuscript would be suitable for publication in ACP if either it is restructured to focus on how the methods used are unique and different from past work or if the authors investigate longer periods to generate more robust analysis and conclusions. Please find my major and minor comments below.*

**Response:**

Thanks to the reviewer for the valuable comments and suggestions which are very helpful for us to improve our manuscript. We have revised the manuscript carefully, as described in our point-to-point responses to the comments.

The major innovation of this study is that it is the first time to quantify the respective/combined contributions of aerosol-photolysis interaction (API) and aerosol-radiation feedback (ARF) on $O_3$ concentrations during multi-pollutant air pollution episodes characterized by high $O_3$ and $PM_{2.5}$ levels. According to the reviewer's comments, another two complex air pollution episodes are also analyzed for generating general conclusions, and we find that API is the dominant factor for $O_3$ reduction related to aerosol-radiation interactions during all the simulated episodes (Episode 1: 28 July-3 August 2014; Episode 2: 8-13 July 2015; Episode 3: 5-11 June 2016).

**Major Comments:**

1. *The novelty of this study is that it is the first time that API and ARF are investigated for synchronous occurrences of high $PM_{2.5}$ and $O_3$ concentrations. This is a rather broad research question to be focused on only one region and one very minor time period. Why do the authors not conduct simulations for either several of these small pollution episodes in this region or for similar episodes in other locations in China?*

**Response:**

The high-resolution WRF-Chem model has been widely applied to investigate the evolution mechanisms of air pollutants during short time periods (Gao et al., 2016; Qiu et al., 2017; Gao et al., 2018; Wang et al., 2020). Gao et al. (2016) summarized the general conclusion that haze events were mainly caused by high emissions of air pollutants and unfavorable weather conditions in North China Plain (NCP) by analyzing a simulated pollution episode from WRF-Chem during 14-24 January 2010. According to the results from WRF-Chem, Qiu et al. (2017) reported that the direct radiative effects of scattering aerosols were greater than that of absorbing aerosols in NCP during 21-27 February 2014. Gao et al. (2018) found that the interactions between black carbon and planetary boundary layer (PBL) could influence the surface $O_3$ concentration in Nanjing during 17 October 2015 by using the process analysis in WRF-Chem.

According to the reviewer's comments, another two complex air pollution episodes (8-13 July 2015 and 5-11 June 2016) in this region are also selected to conduct simulations for generating general conclusions.

[Figure]

**Figure R1.** Changes in surface-layer ozone due to (a1-a2) aerosol-photolysis interaction (API), (b1-b2) aerosol-radiation feedback (ARF), and (c1-c2) the combined effects (ALL, defined as API+ARF) during the daytime (08:00-17:00 LST) from 8-13 July 2015 (upper) and 5-11 June 2016 (bottom). The region sandwiched between two black lines is defined as the complex air pollution areas (CAPAs) where the mean daily PM$_{2.5}$ and MDA8 $O_3$ concentrations in BASE case are larger than 75 µg m$^{-3}$ and 80 ppb. The calculated changes averaged over CAPAs are also shown at the top of each panel.

Simulated air pollutants ($PM_{2.5}$ and $O_3$) and meteorological variables ($T_2$, $RH_2$, and $WS_{10}$) during 8-13 July 2015 (Episode 2) and 5-11 June 2016 (Episode 3) are compared with observations. In general, both the observed meteorological parameters and pollutant concentrations can be reasonably reproduced by the model, with correlation coefficients (R) of 0.56~0.98 and normalized mean bias (NMB) of −7.1%~+33.4%. More details about the model evaluation are listed in the supporting information (**Text S1**).

As shown in Fig. R1(a1-a2), API alone leads to the decrease in surface $O_3$ over the entire domain with an average reduction of 9.0 ppb (10.6%) and 8.3 ppb (10.4%) over CAPAs in Episode 2 and Episode 3, respectively. The decreased surface $O_3$ concentrations over CAPAs due to ARF are only 1.0 ppb (1.2%, Fig. R1(b1)) and 1.0 ppb (1.1%, Fig. R1(b2)) during Episode 2 and Episode 3, respectively. All the results indicate that API is the dominant factor for $O_3$ reduction related to aerosol-radiation interactions, the same as the conclusion analyzed from the case during 28 July to 3 August 2014 (Episode 1). The combined effects of API and ARF decrease surface $O_3$ by 10.0 ppb (11.9%) and 9.3 ppb (11.6%) over CAPAs in Episode 2 and Episode 3, respectively. (**Page 13-14, Line 343-365**)

According to the reviewer's suggestion, Figure R1 is added in the supporting information (**Figure S9**).

2. *Given that government controls have substantially changed emissions in the last decade and will continue into future, how will this research remain relevant in the future or how relevant is it to today's air pollution in China, since the period examined is 7 years ago?*

**Response:**

The stringent Air Pollution Action Plan has been released by the Chinese government in September 2013 to improve the $PM_{2.5}$ air quality. Although the concentrations of $PM_{2.5}$ are decreasing, the concentrations of $PM_{2.5}$ still exceed 35 $\mu g\ m^{-3}$, and the $O_3$ levels have continued to increase (Dai et al., 2021). Many studies have found that the decreased $PM_{2.5}$ can be one of the important causes leading to the increase in $O_3$ (Li et al. 2019; Shao et al., 2021). Li et al. (2019) pointed out that the concentrations of $PM_{2.5}$ were decreased by 40% in North China Plain from 2013 to 2017, which reduced the sink of $HO_2$ on aerosol surfaces and resulted in the increase in $O_3$ by analyzing simulation results from the GEOS-Chem model. Meanwhile, the concentrations of $O_3$ can also be influenced by aerosol-radiation interactions, including aerosol-photolysis interaction and aerosol-radiation feedback, which have not been systematically analyzed. The quantification of the impacts of aerosols on $O_3$ is important to well understand the co-benefits associated with reductions in both aerosols and $O_3$.

In this study, we investigate the impacts of aerosol-radiation interactions on surface $O_3$, and find that the combined impacts of weakened photolysis rates and changed meteorological conditions reduce surface-layer $O_3$ concentrations by up to 11.4 ppb (13.5%). The result can imply that the decreases in $PM_{2.5}$ can lead to the increase in $O_3$ due to the weakened aerosol-radiation interactions, which indicates that if the government controls the anthropogenic emissions in future by using the same strategy, higher $O_3$ will be observed. The result can further emphasize the importance of tighter controls in $O_3$ precursors (i.g., VOCs) to counteract the increased $O_3$ caused by weakened aerosol-radiation interactions. Therefore, the contributions of different mitigation strategies with the impacts of aerosol-radiation interactions to $O_3$ air quality will be discussed

detailedly in our future work.

3. *Is the focus of this research the method in which API and ARF are investigated or the impact of API and ARF in North China? If it is the former than the authors need to reword the abstract, conclusions, and objectives to make it clear that this study is a "proof-of-concept" study on how to best investigate API and ARF in high $O_3$ and $PM_{2.5}$ episodes. If the focus is the latter, the authors need to do additional simulations of other high multi-pollutant episodes, perhaps some closer to current conditions and others in the mid 2000s to see if there is change over time or to make the analysis and conclusions more robust.*

**Response:**

This study mainly focuses on the impacts of API and ARF in North China. According to the reviewer's comments, another two complex air pollution episodes (8-13 July 2015 and 5-11 June 2016) in this region are also selected to conduct simulations for generating general conclusions. The impacts of API and ARF on $O_3$ are shown in Fig. R1, and API is the dominant factor for $O_3$ reduction related to aerosol-radiation interactions. Similar results can also be concluded by analyzing the episode during 28 July to 3 August 2014.

4. *Does this version of WRF-Chem's CBM-Z and MOSAIC modules have a volatility basis set (VBS) option to simulate secondary organic aerosols and if so is it used? Given that, this is a high $O_3$ and $PM_{2.5}$ episode there should be a substantial amount of secondary organic aerosol from abundant oxidants and precursors that may be missed in the model without an advanced SOA scheme. How do the author's address the impact of SOA on their conclusions?*

**Response:**

The selected gas-phase chemical mechanism (CBM-Z) and the aerosol model (MOSAIC) in this study do not consider the impacts of secondary organic aerosols (SOA). The same schemes have been widely used in many other studies, which mainly focus on the impacts of aerosol-radiation interactions on air pollutants in North China (Ding et al., 2016; Gao et al., 2016; Qiu et al., 2017; Chen et al., 2019; Zhou et al, 2019; Gao et al., 2020).

Thanks for the reviewer's suggestion, and we will consider the impacts of SOA in our future works. A discussion about the impacts of SOA has been added in the revised manuscript as follows: "Gao et al. (2017) added some SOA formation mechanisms into the MOSAIC module by using the volatility basis set (VBS) in WRF-Chem and found that the surface $PM_{2.5}$ concentrations in urban Beijing were reduced by 1.9 µg m$^{-3}$ due to the weakened ARF effect during Asia-Pacific Economic Cooperation (APEC). Similar magnitude can also be found in Zhou et al. (2019) (-1.8 µg m$^{-3}$) who did not consider the impacts of SOA in WRF-Chem when analyzing the impacts of weakened ARF on $PM_{2.5}$ during APEC. Therefore, more work should be conducted to explore the impacts of ARF on $PM_{2.5}$ and $O_3$ concentrations under consideration of SOA in future." (**Page 15, Line 402-412**)

5. *The authors are investigating aerosol radiation interactions, but the authors do not evaluate the model's performance against either radiation balance datasets or aerosol optical depth. Since these parameters are more important than surface*

*evaluations of air pollutants to understanding API and ARF, the authors should evaluate their model configuration against satellite AOD and radiation variables such as MODIS or CERES-EBAF.*

**Response:**

[Figure]

**Figure R2.** Spatial distributions of aerosol optical depth (AOD) at 550 nm retrieved from MODIS (left) and simulated by WRF-Chem (right) during 28 July to 3 August 2014. The MODIS retrievals are a combination of the standard (over ocean) and "Deep Blue" (over land) products.

Figure R2 shows the spatial distributions of aerosol optical depth (AOD) at 550 nm retrieved from MODIS and simulated by WRF-Chem during 28 July to 3 August 2014. In the WRF-Chem model, the AOD at 550 nm are calculated by using the values at 400 and 600 nm according to the Ångstrom exponent. Analyzing Fig. R2, the model can well reproduce the spatial distribution of observed AOD but slightly underestimate the value. The spatial correlation coefficient between the simulated and observed AOD is 0.98.

According to the reviewer's suggestion, the description of the model evaluation between observed and simulated AOD is added in the revised manuscript (**Page 8, Line 190-196**), and Figure R2 is also added in the supporting information (**Figure S1**).

6. *Are there only three meteorological observation stations in the domain against? If so, why do the authors not also validate their meteorological performance against gridded products like the Climate Research Unit (CRU) datasets to ensure their performance statistics are robust?*

**Response:**

Thanks to the reviewer's comments. More meteorological observations in the analyzed domain (Table R1) have been used to validate the model results, and the locations of each site are shown in Fig. R3.

Figure R4 shows the time series of observed and simulated $T_2$, $RH_2$, $WS_{10}$ and $J[NO_2]$ during 28 July to 3 August 2014. The observed $T_2$, $RH_2$ and $WS_{10}$ are averaged from the ten meteorological observation stations, and the photolysis rates of $NO_2$ are collected from Peking University. Generally, the WRF-Chem model can depict the temporal variation of $T_2$ fairly well

with R of 0.98 and the mean bias (MB) of -1.5 °C. For $RH_2$, the R and MB are 0.91 and 0.5%, respectively. Although WRF-Chem overestimates $WS_{10}$ with the MB of 0.7 m s$^{-1}$, the root-mean-square error (RMSE) is 0.9 m s$^{-1}$, which is smaller than the threshold of model performance criteria (2 m s$^{-1}$) proposed by Emery et al. (2001). The predicted $J[NO_2]$ agrees well with the observations with R of 0.97 and NMB of 6.8%.

According to the reviewer's comments, we have modified the model evaluation in the revised manuscript (**Page 8, Line 198-208**).

The gridded products like the Climate Research Unit (CRU) datasets covers a large area and a longtime period, which aims to improve scientific understanding of the climate system and its interactions with society. However, the spatial (0.5° × 0.5°) and temporal (monthly) resolution may be too coarse to validate the model performance for generating robust results.

**Table R1.** Locations of the ten stations from NOAA's National Climatic Data Center used in this study.

| Station | Latitude (°) | Longitude (°) |
|---|---|---|
| Yuxian | 39.833 | 114.567 |
| Fengning | 41.2 | 116.633 |
| Zhangjiakou | 40.783 | 114.883 |
| Huailai | 40.417 | 115.5 |
| Chengde | 40.967 | 117.917 |
| Beijing | 40.08 | 116.585 |
| Tianjin | 39.1 | 117.167 |
| Binhai | 39.124 | 117.346 |
| Tangshan | 39.65 | 118.1 |
| Baoding | 38.733 | 115.483 |

[Figure]

**Figure R3.** Map of the two WRF-Chem modeling domains with the locations of meteorological (white dots) and environmental (red crosses) observation sites used for model evaluation.

[Figure]

**Figure R4.** Time series of 3-hourly observed (blue dots) and hourly simulated (red lines) (a) 2-m temperature ($T_2$), (b) 2-m relative humidity ($RH_2$), (c) wind speed at 10 m ($WS_{10}$) averaged over ten meteorological observation stations, and (d) surface photolysis rate of $NO_2$ ($J[NO_2]$) during 28 July to 3 August 2014. The calculated correlation coefficient (R), mean bias (MB), and normalized mean bias (NMB) are also shown.

7. *Given that interactions between $O_3$ and $PM_{2.5}$ are non-linear, how do the authors justify using a simple ratio value (i.e., ROP) to relate these interactions? If this ratio does not account for non-linearity, how useful is this value?*

**Response:**

Thanks to the reviewer's suggestion. As the relationship between $O_3$ and $PM_{2.5}$ is non-linear, and the simple index of ROP can not fully represent the impacts of aerosols on surface $O_3$, so we delete the ROP in the revised manuscript.

8. *The axis labels and legends of Figure 7 are difficult to read. Either each panel should be larger overall or the font sizes of the axes and legends need increased.*

**Response:**

According to the reviewer's suggestion, we have modified the axis labels and legends of Figure 7 and the other figures in the revised manuscript. (**Page 32**)

**Minor Comments:**

1) *In the abstract, there is no context for the values listed. Further reading into the manuscript reveals that these values are the averages in the areas of the complex air pollution areas. The authors should briefly state that these values are for daytime average changes in complex air pollution areas in the*

*abstract. I would also suggest adding a more processed based explanation of the changes in atmospheric state rather than simply listing a long series of values. For example, the authors could state something similar to the following: "Aerosol radiation interactions lead to shortwave dimming at the earth's surface of X, which reduce photolysis rates by X. The dimming stabilizes the atmosphere via surface cooling of X, which reduces PBL height by X. The stabilized atmosphere increases saturation in the lower atmosphere by X. etc…."*

**Response:**

According to the reviewer's suggestion, we have revised the explanation in the abstract as follows: "Our results show that aerosol-radiation interactions decreased the daytime shortwave radiation at surface by 93.2 W m$^{-2}$ averaged over the complex air pollution areas. The dimming effect reduced the 2 m temperature and near-surface photolysis rates of J[NO$_2$] and J[O$^1$D] by 0.56 °C, $1.8 \times 10^{-3}$ s$^{-1}$ and $6.1 \times 10^{-6}$ s$^{-1}$, respectively. However, the daytime shortwave radiation in the atmosphere was increased by 72.8 W m$^{-2}$, which made the atmosphere more stable. The stabilized atmosphere decreased the planetary boundary layer height and 10 m wind speed by 129.0 m and 0.12 m s$^{-1}$, respectively, and increased the relative humidity at 2 m by 2.4%." (**Page 2, Line 24-32**)

2) *Make it clear throughout the manuscript when you are referring to surface level O$_3$ and PM$_{2.5}$.*

**Response:**

According to the reviewer's suggestion, we have revised the expressions in the whole manuscript.

3) *Lines 179-181: The missing PM$_{2.5}$ could also be from missing SOA formation pathways, as mentioned above, if no advanced SOA formulations are used.*

**Response:**

Thanks for your suggestion. The selected aerosol model (MOSAIC) in this study does not consider the impacts of secondary organic aerosols (SOA), and we have added the explanation in the revised manuscript as follows: "The failure to reproduce PM$_{2.5}$ peak values may be attributed to incomplete treatments of chemical reactions in WRF-Chem, e.g., missing the heterogeneous chemistry in the model (Cheng et al., 2016) and the lack of secondary organic aerosols (SOA) formation pathways in the aerosol module (Chen et al., 2019)." (**Page 7-8, Line 183-188**)

4) *Is "downward shortwave radiation in the atmosphere" the SWDNT variable from WRF-Chem? If so, the name of this variable is "downward shortwave radiation at the top of the atmosphere".*

**Response:**

Thanks for your comments. In the WRF-Chem model, SWDNT (SWUPT) means the download (upward) shortwave radiation at the top of atmosphere, and SWDNB (SWUPB) represents the download (upward) shortwave radiation at the surface. According to Zhao et al. (2011), the shortwave radiation in the atmosphere (ATM_SW) can be calculated as the

difference between TOP_SW (the net shortwave radiation at the top of atmosphere, i.e., SWDNT minus SWUPT) and BOT_SW (the net shortwave radiation at the surface, i.e., SWDNB minus SWUPB).

According to the reviewer's suggestion, we have changed the expressions of BOT_SW (shortwave radiation at the surface) and ATM_SW (shortwave radiation in the atmosphere) in the whole revised manuscript.

5) *Lines 217-218: If ATM_SW is the SWDNT variable, what is causing it to increase? SWDNT is usually controlled by the solar constant. Is it possible this is reflected upward shortwave (SWUPT)?*

**Response:**

ATM_SW represents the shortwave radiation in the atmosphere, and it can be calculated by the following equation: ATM_SW = (SWDNT - SWUPT) – (SWDNB - SWUPB).

6) *Lines 248-249: This should be revised to make it clearer that ARF primarily impacts $O_3$ through changing the NOx distribution.*

**Response:**

According to the comments of Reviewer#1, we have deleted this sentence.

7) *Lines 270-281: Is VMIX increasing surface $O_3$ because it is mixing down higher $O_3$ concentrations from aloft or because vertical mixing is suppressed due to a stable atmosphere?*

**Response:**

VMIX increases the surface $O_3$ concentrations by transporting the higher $O_3$ from aloft to the surface layer. Similar results can also be found in previous studies (Tang et al., 2017; Xing et al., 2017; Gao et al., 2018).

8) *Lines 282-294: Why does the VMIX contribution increase because of API?*

**Response:**

Analyzing the vertical profiles of the differences in contributions from each physical/chemical process to hourly $O_3$ variations caused by API in Fig. 8(b), we found that the contribution of VMIX_DIF is negative in the aloft (among the 9th and the 13th layers), while it turns to be positive at the lower seven layers, and the positive contribution increases as the height decreases. The positive variation in VMIX due to API may be associated with the different vertical gradient of $O_3$ between BASE and NOAPI cases.

Similar results can also be found in Gao et al. (2020), who concluded that the increased vertical gradients of $O_3$ due to API could enhance the vertical entrainment.

9) *Lines 295-301: Explain why VMIX_DIF and CHEM_DIF are positive during the day due to ARF.*

**Response:**

When the impacts of ARF are considered, PBLH is decreased over CAPAs (Fig. S3(b3)), which indicates that the suppressed PBL in NOAPI restrains the vertical turbulence and prevents $O_3$ being transported from aloft to surface, resulting in lower $O_3$ concentrations at

surface when comparing with the simulation results of NOALL. However, as the evolution in boundary layer during the daytime, more $O_3$ can be diffused from the upper layers to the surface in NOAPI, and the differences in hourly variation in surface $O_3$ due to vertical mixing between NOAPI and NOALL are positive. Similar results can also be found in Gao et al. (2018).

The typical VOCs/$NO_x$ ratio is calculated to classify sensitivity regimes and to indicate the possible $O_3$ responses to changes in VOCs and/or $NO_x$ concentrations. $O_3$ production is VOC-limited if the ratio is less than 4, and it is $NO_x$-limited if the ratio is larger than 15 (Edson et al., 2017; Li et al., 2017). The ratio of VOCs/$NO_x$ ranging around 4-15 indicates a transitional regime, where ozone is nearly equally sensitive to each species (Sillman, 1999). As shown in Fig R5(a-c), $O_3$ are mainly formed under the VOC-limited and the transition regimes in CAPAs, which means that the increased concentrations of VOCs and $NO_x$ are favorable for ozone chemical production. As shown in Fig. R5(e) and (h), both the surface concentrations of VOCs and $NO_x$ are increased when the impacts of ARF are considered. Thus, the contribution of CHEM in NOAPI is larger than that in NOALL. Similar results can also be found in Gao et al. (2018).

[Figure]

[Figure]

[Figure]

**Figure R5.** The ratios of VOCs/NO$_x$ calculated from (a) BASE, (b) NOALL, and (c) NOAPI. The changed surface-layer concentrations of VOCs and NO$_x$ (NO$_2$+NO, ppb) caused by (d, g) API, (e, h) ARF, and (f, i) ALL during the daytime (08:00-17:00 LST) from 28 July to 3 August 2014. The calculated values averaged over CAPAs are also shown at the top of each panel.

10) *Lines 315-316: Explain how different vertical O$_3$ gradients can cause this change.*

**Response:**

Since the VMIX is closely dependent on atmospheric turbulence and vertical gradients of O$_3$ concentration. The API will increase vertical gradients of O$_3$ to enhance the vertical entrainment (Gao et al., 2020).

Line Comments:

1) *Line 49: This should be "Earth's radiative balance" or "Earth's energy balance"*

**Response:**

Thanks for your suggestion. We have changed the expression in the revised manuscript. (**Page 3, Line 48**)

2) *Lines 54-56: Are these studies all focused on China? If so, state that in the sentence. Change "were" to "are".*

**Response:**

According to the reviewer's suggestion, we have changed the expression in the revised manuscript. (**Page 3, Line 54**)

3) *Lines 56-63: State the domain and time period of Gao et al., (2015) at the beginning of this statement rather than the end*

**Response:**

According to the reviewer's suggestion, we have changed the expression in the revised manuscript. (**Page 3, Line 55-62**)

4) *Line 66: Add "the" before North China Plain*

**Response:**

Thanks for your suggestion. We have added the "the" before North China Plain in the revised manuscript. (**Page 3, Line 64**)

5) *Lines 66-67: If this is referring to surface PM$_{2.5}$ concentrations, add "surface" before PM$_{2.5}$ concentrations.*

**Response:**

Thanks for your suggestion. We have added the "surface" before PM$_{2.5}$ concentrations in the revised manuscript. (**Page 3, Line 65**)

6) *Line 204: should be "attention"*

**Response:**

Thanks for your suggestion. We have changed the expression in the revised manuscript. (**Page 9, Line 218**)

7) *Line 256: Center align the equation.*
**Response:**
This equation has been deleted.

8) *Line: 259: Why are there parentheses in the units?*
**Response:**
This sentence has been deleted.

9) *Lines 288-289: This sentence is a little confusing. Is Net_DIF the sum of CHEM_DIF, VMIX_DIF, and ADV_DIF? If so, state that explicitly and then indicate what Net_DIF describes.*
**Response:**
Thanks for your suggestion. We have defined the NET_DIF in the revised manuscript. (**Page 11, Line 291-292**)

10) *Line 321: Remove "in the"*
**Response:**
According to the reviewer's suggestion, we have deleted it in the revised manuscript.

11) *Line 361: Remove "the contribution from VMIX and"*
**Response:**
According to the reviewer's suggestion, we have deleted it in the revised manuscript.

12) *Line 373: Either "A recent study" or "Recent studies have"*
**Response:**
According to the reviewer's suggestion, we have changed the expression in the revised manuscript. (**Page 15, Line 396**)

**Thank you very much for your comments and suggestions.**

---

## Author Comment (AC3)

**Response to Comments of Reviewer #1**

**(comments in *italics*)**

**Manuscript number:** acp-2021-119

**Title:** Impacts of aerosol-photolysis interaction and aerosol-radiation feedback on surface-layer ozone in North China during a multi-pollutant air pollution episode

*Yang et al. examined the impacts of aerosols on surface ozone through the two well-known pathways, i.e., aerosol-photolysis interaction and aerosol-radiation feedback. The novelty of this study is its focus on the polluted episodes with elevated both PM$_{2.5}$ and ozone levels over North China. They also quantified the chemical and physical processes that drive the aerosol-radiation interactions.*

*Overall, this is a timely study and it clearly demonstrates the impacts of aerosols on ozone pollution. The structure of this manuscript is easy to follow. Although some of the manuscript needs further clarification, the results are generally convincing. As such, I think it is publishable after the following issues are addressed.*

**Response:**

Thanks to the reviewer for the valuable comments and suggestions which are very helpful for us to improve our manuscript. We have revised the manuscript carefully, as described in our point-to-point responses to the comments.

**Specific Comments:**

*1. In Abstract: ozone changes refer to MDA8 ozone or daytime ozone?*

**Response:**

The ozone changes in abstract mean daytime ozone. According to the reviewer's comments, we have added this information in the revised manuscript. (**Page 2, Line 34**)

*2. Line 177: a correlation coefficient of 0.66 reads like not high!*

**Response:**

According to the comments of Reviewer#2, another two complex air pollution episodes (8-13 July 2015 (Episode2) and 5-11 June 2016 (Episode3)) in this region are selected to conduct simulations for generating general conclusions.

Thanks for your suggestion, we have changed this sentence in the revised manuscript as follows: "As shown in Fig. 2, the temporal variations of observed PM$_{2.5}$ can be well performed by the model with correlation coefficients (R) of 0.66, 0.56 and 0.73 and normalized mean bias (NMB) of -19.2%, -3.9% and 30.4% during Episode1, Episode2 and Episode3, respectively." (**Page 7-8, Line 183-186**)

*3. Lines 179-181: the oxidation of SO$_2$ by NO$_2$ in aqueous aerosols is important for summertime?*

**Response:**

This sentence has been deleted in the revised manuscript.

4. *Lines 248-251: this statement looks reasonable here, but in the later text the process analysis shows that chemistry will be enhanced by ARF. Instead, ARF decreases ozone through physical processes.*

**Response:**

   Thanks for the reviewer's suggestion. We have deleted this sentence in the revised manuscript.

5. *Line 260: "is" should be "are". Please do proof-reading throughout the text.*

**Response:**

   This sentence has been deleted in the revised manuscript. According to the reviewer's comments, proof-reading has been conducted through the whole revised manuscript.

6. *Line 310: It is Okay to use model levels (e.g., 12 levels), but it will be better to add model height in meters as well.*

**Response:**

   Thanks for reviewer's suggestion. We have added the model height in meters in the revised manuscript. (**Page 12-13, Line 320, Line 323-325, Line 334-335**)

7. *Lines 326-327: why do you need this statement?*

**Response:**

   Analyzing Fig. 8c we can conclude that ARF promotes the $O_3$ chemical production with a positive mean value of 0.72 ppb $h^{-1}$. The enhanced $O_3$ precursors due to ARF can promote the chemical production of $O_3$. According to the reviewer's comment, we have deleted this statement in the revised manuscript.

8. *Lines 327-328: Please provide evidence to support this conclusion.*

**Response:**

   The typical VOCs/$NO_x$ ratio is calculated to classify sensitivity regimes and to indicate the possible $O_3$ responses to changes in VOCs and/or $NO_x$ concentrations. $O_3$ production is VOC-limited if the ratio is less than 4, and it is $NO_x$-limited if the ratio is larger than 15 (Edson et al., 2017; Li et al., 2017). The ratio of VOCs/$NO_x$ ranging around 4-15 indicates a transitional regime, where ozone is nearly equally sensitive to each species (Sillman, 1999). As shown in Fig. R1(a-f), $O_3$ are mainly formed under VOC-limited and transition regimes in CAPAs, which means that the increased concentrations of VOCs and $NO_x$ are favorable for ozone chemical production. As shown in Fig. R1(g-i) and (j-l), both the surface concentrations of VOCs and $NO_x$ are increased when the impacts of ARF are considered. Thus, the contribution of CHEM in NOAPI is larger than that in NOALL. Similar results can also be found in Gao et al. (2018).

[Figure]

**Figure R1.** The ratios of VOCs/$NO_x$ calculated from (a-c) NOALL, and (d-f) NOAPI. The changed surface-layer concentrations of (g-i) VOCs and (j-l) $NO_x$ ($NO_2$+NO, ppb) caused by ARF during the daytime (08:00-17:00 LST) from Episode1 to Episode3. The calculated values averaged over CAPAs are also shown at the top of each panel.

9. *Discussion. I think the authors should do some comparisons between your results with previous studies. This is important for readers to better understand your case study results. Moreover, how about the applicability of the calculated ROP of -0.14*

*ppb (µg m$^{-3}$)$^{-1}$?*
**Response:**

According to the comments of Reviewer#2, we conduct another two complex air pollution episodes (8-13 July 2015 and 5-11 June 2016) in this region to draw the general conclusions. The three episodes feature a similar variation pattern, and the detailed information can be found in section 4. (**Page 9-13, Line 223-352**). Meanwhile, a discussion about the impacts of secondary organic aerosols (SOA) is also added in the section 5 (**Page 14-15, Line 391-401**).

Thanks to the reviewer's comments. As the relationship between $O_3$ and $PM_{2.5}$ is non-linear, and the simple index of ROP can not fully represent the impacts of aerosols on surface $O_3$, so we delete the ROP in the revised manuscript.

*10. Fig.2: It will be better to add error bars for observed $PM_{2.5}$ and ozone.*
**Response:**

According to the reviewer's suggestion, error bars have been added in Fig. 2 in the revised manuscript. (**Page 27**)

*11. Fig.3: what are the cities these plots for?*
**Response:**

The averaged $T_2$, $RH_2$, and $WS_{10}$ are collected from ten meteorological observation stations, and the detail information about the sites is listed in Table S1. The photolysis rates of $NO_2$ ($J[NO_2]$) are observed in Peking University. More details are explained in section 2.3. (**Page 6**)

*12. Fig.7: what are the layers your process analysis applied for? I don't see this key information here, as well as in the text.*
**Response:**

The surface-layer, namely, first-layer $O_3$ concentrations are analyzed in Fig. 7. Thanks for reviewer's suggestion, we have added this information in the revised manuscript. (**Page 10, Line 272**)

**Reference:**

Edson, C. T., Ivan, H.-P. and Alberto, M.: Use of combined observational- and model-derived photochemical indicators to assess the $O_3$-$NO_x$-VOC System sensitivity in urban areas, Atmosphere., 8, 22. https://doi.org/10.3390/ atmos8020022, 2017.

Gao, J. H., Zhu, B., Xiao, H., Kang, H. Q., Pan, C., Wang, D. D., and Wang, H. L.: Effects of black carbon and boundary layer interaction on surface ozone in Nanjing, China, Atmos. Chem. Phys., 18, 7081–7094, https://doi.org/10.5194/acp-18-7081-2018, 2018.

Li, K., Chen, L., Ying, F., White, S. J., Jang, C., Wu, X., Gao, X., Hong, S., Shen, J., Azzi, M. and Cen, K: Meteorological and chemical impacts on ozone formation: a case study in Hangzhou, China, Atmos. Res., 196, https://doi.org/10.1016/ j. atmosres.2017.06.003, 2017.

Sillman, S.: The relation between ozone, $NO_x$ and hydrocarbons in urban and polluted rural environments, Atmos. Environ., 33, 1821-1845, https://doi.org/ 10.1016/S1352-2310(98)00345-8, 1999.

**Thank you very much for your comments and suggestions.**

---

## Author Comment (AC4)

**Response to Comments of Reviewer #2**

**(comments in *italics*)**

**Manuscript number: acp-2021-119**

**Title:** Impacts of aerosol-photolysis interaction and aerosol-radiation feedback on surface-layer ozone in North China during a multi-pollutant air pollution episode

In this study, Yang et al. investigate the impact of aerosol-radiation interactions on O3 formation during a multi-pollutant air pollution episode in Northern China. Additionally, the study uses process analysis to analyze how the aerosol-radiation interactions affect O3 through various physical and chemical mechanisms. This is an interesting research topic with valid research methods and an overall well written and well-structured manuscript. However, the period of analysis is far too short (i.e., 7 days) to robustly quantify the impact of aerosol-radiation impacts in this region or to describe any variability. Additionally, the time period analyzed appears somewhat arbitrary and is nearly a decade removed from current conditions. For these reasons, the manuscript is not currently at the scientific level of the Atmospheric Chemistry and Physics Journal. However, this manuscript would be suitable for publication in ACP if either it is restructured to focus on how the methods used are unique and different from past work or if the authors investigate longer periods to generate more robust analysis and conclusions. Please find my major and minor comments below.

**Response:**

Thanks to the reviewer for the valuable comments and suggestions which are very helpful for us to improve our manuscript. We have revised the manuscript carefully, as described in our pointto-point responses to the comments.

The major innovation of this study is that it is the first time to quantify the respective/combined contributions of aerosol-photolysis interaction (API) and aerosol-radiation feedback (ARF) on  $O_3$  concentrations during multi-pollutant air pollution episodes characterized by high  $O_3$  and  $PM_{2.5}$  levels. According to the reviewer's comments, another two complex air pollution episodes are also analyzed for generating general conclusions, and we find that API is the dominant factor for  $O_3$  reduction related to aerosol-radiation interactions during all the simulated episodes (Episode1: 28 July-3 August 2014; Episode2: 8-13 July 2015; Episode3: 5-11 June 2016).

**Major Comments:**

1. The novelty of this study is that it is the first time that API and ARF are investigated for synchronous occurrences of high  $PM_{2.5}$  and  $O_3$  concentrations. This is a rather broad research question to be focused on only one region and one very minor time period. Why do the authors not conduct simulations for either several of these small pollution episodes in this region or for similar episodes in other locations in China?

**Response:**

The high-resolution WRF-Chem model has been widely applied to investigate the evolution mechanisms of air pollutants during short time periods (Gao et al., 2016; Qiu et al., 2017; Gao et

al., 2018; Wang et al., 2020). Gao et al. (2016) summarized the general conclusion that haze events were mainly caused by high emissions of air pollutants and unfavorable weather conditions in North China Plain (NCP) by analyzing a simulated pollution episode from WRF-Chem during 14-24 January 2010. According to the results from WRF-Chem, Qiu et al. (2017) reported that the direct radiative effects of scattering aerosols were greater than that of absorbing aerosols in NCP during 21-27 February 2014. Gao et al. (2018) found that the interactions between black carbon and planetary boundary layer (PBL) could influence the surface O3 concentration in Nanjing during 17 October 2015 by using the process analysis in WRF-Chem.

According to the reviewer's comments, another two complex air pollution episodes (8-13 July 2015 and 5-11 June 2016) in this region are also selected to conduct simulations for generating general conclusions.

**Figure R1.** The changes in surface-layer ozone due to (a) aerosol-photolysis interaction (API), (b) aerosol-radiation feedback (ARF), and (c) the combined effects (ALL, defined as API+ARF) in the daytime (08:00-17:00 LST) during 28 July to 3 August 2014 (Episode1), 8-13 July 2015 (Episode2) and 5-11 June 2016 (Episode3). The region sandwiched between two black lines is defined as the complex air pollution areas (CAPAs) where the mean daily

 $PM_{2.5}$  and MDA8  $O_3$  concentrations in BASE case are larger than 75  $\mu$ g m-3 and 80 ppb. The calculated mean changes (percentage changes) avaraged over CAPAs are also shown at the top of each panel.

Simulated air pollutants (PM2.5 and O3) and meteorological variables (T2, RH2, WS10, and J[NO2]) during 28 July to 3 August 2014 (Episode1), 8-13 July 2015 (Episode2) and 5-11 June 2016 (Episode3) are compared with observations. In general, both the observed meteorological parameters and pollutant concentrations can be reasonably reproduced by the model, with correlation coefficients (R) of  $0.56 \sim 0.98$  and normalized mean bias (NMB) of  $-12.0\% \sim +33.4\%$ . More details about the model evaluation are listed in the section 3 in the revised manuscript (**Page 7-8, Line 181-212**). The impacts of aerosol radiation effects on meteorological variables can be found in section 4.1 and 4.2 in the revised manuscript during these three episodes (**Page 9-10, Line 223-254**).

As shown in Fig. R1(a1-a3), API alone leads to overall surface O3 decreases over the entire domain with average reductions of 8.5 ppb (10.1%), 9.0 ppb (10.6%) and 8.3 ppb (10.4%) over CAPAs in the three episodes, respectively. The changes can be explained by the substantially diminished UV radiation due to aerosol loading, which significantly weakens the efficiency of photochemical reactions and restrains O3 formation. However, the decreased surface O3 concentrations due to ARF are only 2.9 ppb (3.1%, Fig. R1(b1)), 1.0 ppb (1.2%, Fig. R1(b2)) and 1.0 ppb (1.1%, Fig. R1(b3)) for the three episodes, which indicates that API is the dominant way for O3 reduction related to aerosol-radiation interactions. Fig. R1(c1-c3) presents the combined effects of API and ARF. Generally, aerosol-radiation interactions decrease the surface O3 concentrations by 11.4 ppb (13.5%), 10.0 ppb (11.9%) and 9.3 ppb (11.6%) averaged over CAPAs in the three episodes, respectively. (Page 10, Line 256-269)

2. Given that government controls have substantially changed emissions in the last decade and will continue into future, how will this research remain relevant in the future or how relevant is it to today's air pollution in China, since the period examined is 7 years ago?

**Response:**

The stringent Air Pollution Action Plan has been released by the Chinese government in September 2013 to improve the  $PM_{2.5}$  air quality. Although the concentrations of  $PM_{2.5}$  are decreasing, the concentrations of  $PM_{2.5}$  still exceed 35 µg m-3, and the O3 levels have continued to increase (Dai et al., 2021). Many studies have found that the decreased  $PM_{2.5}$  can be one of the important causes leading to the increase in O3 (Li et al. 2019; Shao et al., 2021). Li et al. (2019) pointed out that the concentrations of  $PM_{2.5}$  were decreased by 40% in North China Plain from 2013 to 2017, which reduced the sink of HO2 on aerosol surfaces and resulted in the increase in O3 by analyzing simulation results from the GEOS-Chem model. Meanwhile, the concentrations of O3 can also be influenced by aerosol-radiation interactions, including aerosol-photolysis interaction and aerosol-radiation feedback, which have not been systematically analyzed. The quantification of the impacts of aerosols on O3 is important to well understand the co-benefits associated with reductions in both aerosols and O3.

In this study, we investigate the impacts of aerosol-radiation interactions on surface  $O_3$ , and find that the combined impacts of weakened photolysis rates and changed meteorological conditions reduce surface-layer  $O_3$  concentrations by up to  $9.3 \sim 11.4$  ppb. The result can imply that the

decreases in  $PM_{2.5}$  can lead to the increase in  $O_3$  due to the weakened aerosol-radiation interactions, which indicates that if the government controls the anthropogenic emissions in future by using the same strategy, higher  $O_3$  will be observed. The result can further emphasize the importance of tighter controls in  $O_3$  precursors (e.g., VOCs) to counteract the increased  $O_3$  caused by weakened aerosol-radiation interactions. Therefore, the contributions of different mitigation strategies with the impacts of aerosol-radiation interactions to  $O_3$  air quality will be discussed detailedly in our future work.

3. Is the focus of this research the method in which API and ARF are investigated or the impact of API and ARF in North China? If it is the former than the authors need to reword the abstract, conclusions, and objectives to make it clear that this study is a "proof-of-concept" study on how to best investigate API and ARF in high O3 and PM2.5 episodes. If the focus is the latter, the authors need to do additional simulations of other high multi-pollutant episodes, perhaps some closer to current conditions and others in the mid 2000s to see if there is change over time or to make the analysis and conclusions more robust.

**Response:**

This study mainly focuses on the impacts of API and ARF in North China. According to the reviewer's comments, another two complex air pollution episodes (8-13 July 2015 and 5-11 June 2016) in this region are also selected to conduct simulations for generating general conclusions. The impacts of API and ARF on O3 are shown in Fig. R1, and API is the dominant factor for O3 reduction related to aerosol-radiation interactions in these three episodes.

4. Does this version of WRF-Chem's CBM-Z and MOSAIC modules have a volatility basis set (VBS) option to simulate secondary organic aerosols and if so is it used? Given that, this is a high O3 and PM2.5 episode there should be a substantial amount of secondary organic aerosol from abundant oxidants and precursors that may be missed in the model without an advanced SOA scheme. How do the author's address the impact of SOA on their conclusions?

**Response:**

The selected gas-phase chemical mechanism (CBM-Z) and the aerosol model (MOSAIC) in this study do not consider the impacts of secondary organic aerosols (SOA). The same schemes have been widely used in many other studies, which mainly focus on the impacts of aerosol-radiation interactions on air pollutants in North China (Ding et al., 2016; Gao et al., 2016; Qiu et al., 2017; Chen et al., 2019; Zhou et al, 2019; Gao et al., 2020).

Thanks for the reviewer's suggestion, and we will consider the impacts of SOA in our future works. A discussion about the impacts of SOA has been added in the revised manuscript as follows: "Gao et al. (2017) added some SOA formation mechanisms into the MOSAIC module by using the volatility basis set (VBS) in WRF-Chem and found that the surface  $PM_{2.5}$  concentrations in urban Beijing were reduced by 1.9 µg m-3 due to the weakened ARF effect during Asia-Pacific Economic Cooperation (APEC). Similar magnitude can also be found in Zhou et al. (2019) (-1.8 µg m-3) who did not consider the impacts of SOA in WRF-Chem when analyzing the impacts of weakened ARF on  $PM_{2.5}$  during APEC. Therefore, more work should be conducted to explore the impacts of ARF on  $PM_{2.5}$  and  $O_3$  concentrations under consideration of SOA in future." (**Page 14-15, Line 391-401**)

5. The authors are investigating aerosol radiation interactions, but the authors do not evaluate the model's performance against either radiation balance datasets or aerosol optical depth. Since these parameters are more important than surface evaluations of air pollutants to understanding API and ARF, the authors should evaluate their model configuration against satellite AOD and radiation variables such as MODIS or CERES-EBAF.

**Response:**

---

## Referee Report (RR1)

The revised reversion of Yang et al. does address several of the manuscripts original limitations. The manuscript is still clear and well written and the scientific problem addressed is important to the scientific community. However, a few issues remain that leave questions as to whether the findings in the manuscript are robust and meaningful. My major and minor comments on the revised manuscript are as follows:

Major Comments

1) The authors have included two additional episodes to address the issue of scientific robustness. With these two additional episodes, it appears that the impact of aerosol radiation interactions, usually via API, are similar in all episodes despite variability in the magnitude and spatial extent of the CAPAs. With these finding we can reasonably conclude that these values are indeed representative of CAPAs in this region during the period of 2014-2016. However, this did not address any issues with changes in time/emissions (i.e, 2001-2005 or more currently 2018-2020). If the authors are not going to do any additional episodes, they need to convincingly justify why the period of 2014-2016 is representative of /or important for current/future conditions.

2) The authors have added caveats to the conclusion to address the issue of lacking SOA formation pathways in their simulations. These listed caveats are important, but the authors have overlooked the possibility that increased O3 from PM2.5 reductions will generate more SOA via increased oxidation. This feedback could partially compensate the increased O3 formation the authors predict will happen.

Minor Comments

1) The response to the previous Reviewer2 Minor comments 7-10 should be included in the manuscript if not already done to facilitate ease of understanding.
2) The color bars for Figures 5 and 6 needs to be the same for all episodes to facilitate easy comparison.
3) In Figures 2 and 3 the y-axis need to be consistent for all episodes to facilitate ease of comparison.

---

## Referee Report (RR2)

This revised paper is well written and well reasoned but in previous round of review, several key issues were pointed out. Specifically in the last round of review, the editor posed three key questions that the authors needed to address in the manuscript with which I agree.

1) What is the goal of the paper - to consider high air pollution conditions, and for what purpose?
2) Why were the episodes chosen, and what do we hope to learn more generally for other conditions?
3) How do the conclusions based on the CAPA episodes relate with other conditions during the years studied (2014-16) and to what extent are they relevant for other years before or after?

I believe that the authors have fully answered question 1 and the clear motivation has improved the manuscript. However, the current revision only partially answers questions 2 and 3. My major and minor comments are as follows:

Major Comments:

1a) The authors addressed the questions about other conditions by conducting 3 additional simulations with conditions other than complex air pollution episodes. These simulations are important and reveal some interesting findings. Particularly, the $O_3$ decrease in a high PM episode from API and ARF is close to double the impact in a complex episode and in cases without high PM pollution the impact is close to half the impact of a complex episode. However, the authors treat these cases as an afterthought with them being mentioned only in the last few lines of the conclusion/supplementary material. These cases need to be incorporated into the main text as another results section and they need more descriptive names such as: High $PM_{2.5}$ (HI_PM), Low Pollution (LOW_POL), and High $O_3$ (HI_O3).

1b) The authors should also consider possibly simulating an additional case with each of these conditions to see if the differences between them and the CAPAs are robust, or alternatively they should demonstrate that the episodes they selected are representative of those conditions throughout the period of interest (2014-2017).

2) The authors have still not addressed historical conditions before 2014. The authors have stated this is because national observations are not available to pin-point complex air pollution episodes before 2013. However, the authors have also stated "Air pollution in China was characterized by high concentrations of PM2.5 before 2014 (Li et al., 2019a; Zhang et al., 2019) and by synchronous occurrence of high PM2.5 and $O_3$ or high levels of O3 alone after 2017 (Dai et al., 2021; Li et al., 2019b; Li et al., 2020; Qin et al., 2021)." If there are no observations before 2013 to justify this statement, how is this statement supported? At a minimum, the authors should point out the lack of data before their period of interest in the manuscript as a reason they cannot explore the magnitudes of API and ARF before 2014. However, if the authors are at least aware of a high $PM_{2.5}$ case from the 2001-2005 period it would be worth comparing that with the current HI_PM case to see if the magnitude of the API and ARF impacts have decreased in a significant way.

Minor comments:

Line 78: field not "filed"

Line 84: Should the unit be $\mu g\ m^{-3}$ or ppb?

Line 206: done instead of "did"

Line 291: should be "was caused"

Line 414: How can API contribute more than 100% to a reduction in $O_3$? The word contribution implies "percent of" not "percent change". If these values are a relative difference (i.e., percent change) then a word other than contribute needs to be used.

Line 430: during "the" warm season

---

## Author Response (AR2)

**Response to Comments of Editor**

**(comments in *italics*)**

**Manuscript number:** acp-2021-119

**Title:** Impacts of aerosol-photolysis interaction and aerosol-radiation feedback on surface-layer ozone in North China during a multi-pollutant air pollution episode

*Reviewer #2 said that the paper suffers from analyzing a small region over a few days, making it hard to draw broader conclusions. Reviewer #2 further said that the paper should either 1) make clear how the paper is methdologically novel or 2) analyze a longer period(s) so that we understand better how general the results are. I understand that the focus is on severely polluted conditions, and so you wouldn't simulate a full year, for example.*

*I agree with Reviewer #2's assessment. In response, no case is made for methodological novelty. Two additional short episodes are added to the paper, for which apparently similar conclusions are reached. But those two episodes are added as an after-thought - they appear only in a separate Discussion section, with all relevant figures in the supporting information. No case is made for why these two extra episodes are less important to justify keeping the focus on the first episode.*

*I do not think it would be worthwhile to send the current version to the same referees in its current form. Rather, I think the authors should rewrite the paper to talk about the analysis of 3 episodes throughout, from the introduction through to conclusions - unless the authors can justify the main focus on one episode and secondary focus on the 2 others. You are free to respond as you'd like, but that is my strong recommendation.*

*Apart from that, I felt that the authors gave good responses to many of the individual points from the 2 referees, but often did not go far enough to change the paper itself to address those concerns. For example, Figure R2 seems to me to show that the model is a factor of two too low in AOD.*

*If you would like to resubmit to ACP, please make substantial revisions to the paper to incorporate the 3 episodes throughout the paper, and prepare an improved response to reviewers that more fully shows how the paper is improved as a results of responding to comments.*

**Response:**

Thanks for the Editor's valuable comments to this manuscript. We totally agree with your helpful suggestions, and we have made substantial revisions in the final revised paper.

Three complex air pollution episodes (Episode1: 28 July-3 August 2014, Episode2: 8-13 July 2015, Episode3: 5-11 June 2016) are selected and analyzed throughout the whole revised manuscript, and the general conclusion can be summarized that API (aerosol-photolysis interaction) is the dominant factor for $O_3$ reduction related to aerosol-radiation interactions during all the simulated episodes.

According to the Editor's comments, all the responses to the two reviewers are rewritten. The new updated responses and the final revised paper can clearly show how the manuscript is improved.

**Response to "*Figure R2 seems to me to show that the model is a factor of two too low in AOD*"**

  Thank you for your comments. Previous studies find that MODIS retrievals can overestimate AOD in NCP during polluted events when comparing with observations collected from AERONET (Gao et al., 2015; Li et al., 2016). Therefore, comparisons between simulated AOD and AERONET observations are conducted in this work. The revised Fig. R2 shows the correlation between observed and simulated AOD at 550 nm in Beijing. In the WRF-Chem model, the AOD at 550 nm are calculated by using the values at 400 and 600 nm according to the Angstrom exponent. Analyzing Fig. R2, the model can reproduce the observed AOD with R of 0.7 and NMB of 7.9%. (**Page 8, Line 190-193**)

[Figure]

**Figure R2.** Comparison of observed and simulated AOD at 550 nm in Beijing (39.98°N, 116.38°E). The observed AOD during the three episodes are collected from AERONET.

**Reference:**

Gao, Y., Zhang, M., Liu, Z., Wang, L., Wang, P., Xia, X., Tao, M., and Zhu, L.: Modeling the feedback between aerosol and meteorological variables in the atmospheric boundary layer during a severe fog–haze event over the North China Plain, Atmos. Chem. Phys., 15, 4279–4295, doi:10.5194/acp-15-4279-2015, 2015.

Li, K., Liao, H., Zhu, J., and Moch, J. M.: Implications of RCP emissions on future PM2.5 air quality and direct radiative forcing over China, J. Geophys. Res. Atmos., 121, 12,985–13,008, doi:10.1002/2016JD025623, 2016.

**Thank you very much for your comments and suggestions.**

**Response to Comments of Reviewer #1**

**(comments in *italics*)**

**Manuscript number:** acp-2021-119

**Title:** Impacts of aerosol-photolysis interaction and aerosol-radiation feedback on surface-layer ozone in North China during a multi-pollutant air pollution episode

*Yang et al. examined the impacts of aerosols on surface ozone through the two well-known pathways, i.e., aerosol-photolysis interaction and aerosol-radiation feedback. The novelty of this study is its focus on the polluted episodes with elevated both PM$_{2.5}$ and ozone levels over North China. They also quantified the chemical and physical processes that drive the aerosol-radiation interactions.*

*Overall, this is a timely study and it clearly demonstrates the impacts of aerosols on ozone pollution. The structure of this manuscript is easy to follow. Although some of the manuscript needs further clarification, the results are generally convincing. As such, I think it is publishable after the following issues are addressed.*

**Response:**

Thanks to the reviewer for the valuable comments and suggestions which are very helpful for us to improve our manuscript. We have revised the manuscript carefully, as described in our point-to-point responses to the comments.

**Specific Comments:**

*1. In Abstract: ozone changes refer to MDA8 ozone or daytime ozone?*

**Response:**

The ozone changes in abstract mean daytime ozone. According to the reviewer's comments, we have added this information in the revised manuscript. (**Page 2, Line 34**)

*2. Line 177: a correlation coefficient of 0.66 reads like not high!*

**Response:**

According to the comments of Reviewer#2, another two complex air pollution episodes (8-13 July 2015 (Episode2) and 5-11 June 2016 (Episode3)) in this region are selected to conduct simulations for generating general conclusions.

Thanks for your suggestion, we have changed this sentence in the revised manuscript as follows: "As shown in Fig. 2, the temporal variations of observed PM$_{2.5}$ can be well performed by the model with correlation coefficients (R) of 0.66, 0.56 and 0.73 and normalized mean bias (NMB) of -19.2%, -3.9% and 30.4% during Episode1, Episode2 and Episode3, respectively." (**Page 7-8, Line 183-186**)

*3. Lines 179-181: the oxidation of SO$_2$ by NO$_2$ in aqueous aerosols is important for summertime?*

**Response:**

This sentence has been deleted in the revised manuscript.

4. *Lines 248-251: this statement looks reasonable here, but in the later text the process analysis shows that chemistry will be enhanced by ARF. Instead, ARF decreases ozone through physical processes.*

**Response:**

Thanks for the reviewer's suggestion. We have deleted this sentence in the revised manuscript.

5. *Line 260: "is" should be "are". Please do proof-reading throughout the text.*

**Response:**

This sentence has been deleted in the revised manuscript. According to the reviewer's comments, proof-reading has been conducted through the whole revised manuscript.

6. *Line 310: It is Okay to use model levels (e.g., 12 levels), but it will be better to add model height in meters as well.*

**Response:**

Thanks for reviewer's suggestion. We have added the model height in meters in the revised manuscript. (**Page 12-13, Line 320, Line 323-325, Line 334-335**)

7. *Lines 326-327: why do you need this statement?*

**Response:**

Analyzing Fig. 8c we can conclude that ARF promotes the $O_3$ chemical production with a positive mean value of 0.72 ppb $h^{-1}$. The enhanced $O_3$ precursors due to ARF can promote the chemical production of $O_3$. According to the reviewer's comment, we have deleted this statement in the revised manuscript.

8. *Lines 327-328: Please provide evidence to support this conclusion.*

**Response:**

The typical VOCs/$NO_x$ ratio is calculated to classify sensitivity regimes and to indicate the possible $O_3$ responses to changes in VOCs and/or $NO_x$ concentrations. $O_3$ production is VOC-limited if the ratio is less than 4, and it is $NO_x$-limited if the ratio is larger than 15 (Edson et al., 2017; Li et al., 2017). The ratio of VOCs/$NO_x$ ranging around 4-15 indicates a transitional regime, where ozone is nearly equally sensitive to each species (Sillman, 1999). As shown in Fig. R1(a-f), $O_3$ are mainly formed under VOC-limited and transition regimes in CAPAs, which means that the increased concentrations of VOCs and $NO_x$ are favorable for ozone chemical production. As shown in Fig. R1(g-i) and (j-l), both the surface concentrations of VOCs and $NO_x$ are increased when the impacts of ARF are considered. Thus, the contribution of CHEM in NOAPI is larger than that in NOALL. Similar results can also be found in Gao et al. (2018).

[Figure]

**Figure R1.** The ratios of VOCs/NO$_x$ calculated from (a-c) NOALL, and (d-f) NOAPI. The changed surface-layer concentrations of (g-i) VOCs and (j-l) NO$_x$ (NO$_2$+NO, ppb) caused by ARF during the daytime (08:00-17:00 LST) from Episode1 to Episode3. The calculated values averaged over CAPAs are also shown at the top of each panel.

9. *Discussion. I think the authors should do some comparisons between your results with previous studies. This is important for readers to better understand your case study results. Moreover, how about the applicability of the calculated ROP of -0.14*

*ppb ($\mu g$ $m^{-3}$)$^{-1}$?*

**Response:**

According to the comments of Reviewer#2, we conduct another two complex air pollution episodes (8-13 July 2015 and 5-11 June 2016) in this region to draw the general conclusions. The three episodes feature a similar variation pattern, and the detailed information can be found in section 4. (**Page 9-13, Line 223-352**). Meanwhile, a discussion about the impacts of secondary organic aerosols (SOA) is also added in the section 5 (**Page 14-15, Line 391-401**).

Thanks to the reviewer's comments. As the relationship between $O_3$ and $PM_{2.5}$ is non-linear, and the simple index of ROP can not fully represent the impacts of aerosols on surface $O_3$, so we delete the ROP in the revised manuscript.

*10. Fig.2: It will be better to add error bars for observed $PM_{2.5}$ and ozone.*

**Response:**

According to the reviewer's suggestion, error bars have been added in Fig. 2 in the revised manuscript. (**Page 27**)

*11. Fig.3: what are the cities these plots for?*

**Response:**

The averaged $T_2$, $RH_2$, and $WS_{10}$ are collected from ten meteorological observation stations, and the detail information about the sites is listed in Table S1. The photolysis rates of $NO_2$ ($J[NO_2]$) are observed in Peking University. More details are explained in section 2.3. (**Page 6**)

*12. Fig.7: what are the layers your process analysis applied for? I don't see this key information here, as well as in the text.*

**Response:**

The surface-layer, namely, first-layer $O_3$ concentrations are analyzed in Fig. 7. Thanks for reviewer's suggestion, we have added this information in the revised manuscript. (**Page 10, Line 272**)

**Reference:**

Edson, C. T., Ivan, H.-P. and Alberto, M.: Use of combined observational- and model-derived photochemical indicators to assess the $O_3$-$NO_x$-VOC System sensitivity in urban areas, Atmosphere., 8, 22. https://doi.org/10.3390/ atmos8020022, 2017.

Gao, J. H., Zhu, B., Xiao, H., Kang, H. Q., Pan, C., Wang, D. D., and Wang, H. L.: Effects of black carbon and boundary layer interaction on surface ozone in Nanjing, China, Atmos. Chem. Phys., 18, 7081–7094, https://doi.org/10.5194/acp-18-7081-2018, 2018.

Li, K., Chen, L., Ying, F., White, S. J., Jang, C., Wu, X., Gao, X., Hong, S., Shen, J., Azzi, M. and Cen, K: Meteorological and chemical impacts on ozone formation: a case study in Hangzhou, China, Atmos. Res., 196, https://doi.org/10.1016/ j. atmosres.2017.06.003, 2017.

Sillman, S.: The relation between ozone, $NO_x$ and hydrocarbons in urban and polluted rural environments, Atmos. Environ., 33, 1821-1845, https://doi.org/ 10.1016/S1352-2310(98)00345-8, 1999.

**Thank you very much for your comments and suggestions.**

**Response to Comments of Reviewer #2**

**(comments in *italics*)**

**Manuscript number:** acp-2021-119

**Title:** Impacts of aerosol-photolysis interaction and aerosol-radiation feedback on surface-layer ozone in North China during a multi-pollutant air pollution episode

*In this study, Yang et al. investigate the impact of aerosol-radiation interactions on $O_3$ formation during a multi-pollutant air pollution episode in Northern China. Additionally, the study uses process analysis to analyze how the aerosol-radiation interactions affect $O_3$ through various physical and chemical mechanisms. This is an interesting research topic with valid research methods and an overall well written and well-structured manuscript. However, the period of analysis is far too short (i.e., 7 days) to robustly quantify the impact of aerosol-radiation impacts in this region or to describe any variability. Additionally, the time period analyzed appears somewhat arbitrary and is nearly a decade removed from current conditions. For these reasons, the manuscript is not currently at the scientific level of the Atmospheric Chemistry and Physics Journal. However, this manuscript would be suitable for publication in ACP if either it is restructured to focus on how the methods used are unique and different from past work or if the authors investigate longer periods to generate more robust analysis and conclusions. Please find my major and minor comments below.*

**Response:**

Thanks to the reviewer for the valuable comments and suggestions which are very helpful for us to improve our manuscript. We have revised the manuscript carefully, as described in our point-to-point responses to the comments.

The major innovation of this study is that it is the first time to quantify the respective/combined contributions of aerosol-photolysis interaction (API) and aerosol-radiation feedback (ARF) on $O_3$ concentrations during multi-pollutant air pollution episodes characterized by high $O_3$ and $PM_{2.5}$ levels. According to the reviewer's comments, another two complex air pollution episodes are also analyzed for generating general conclusions, and we find that API is the dominant factor for $O_3$ reduction related to aerosol-radiation interactions during all the simulated episodes (Episode1: 28 July-3 August 2014; Episode2: 8-13 July 2015; Episode3: 5-11 June 2016).

**Major Comments:**

1. *The novelty of this study is that it is the first time that API and ARF are investigated for synchronous occurrences of high $PM_{2.5}$ and $O_3$ concentrations. This is a rather broad research question to be focused on only one region and one very minor time period. Why do the authors not conduct simulations for either several of these small pollution episodes in this region or for similar episodes in other locations in China?*

**Response:**

The high-resolution WRF-Chem model has been widely applied to investigate the evolution mechanisms of air pollutants during short time periods (Gao et al., 2016; Qiu et al., 2017; Gao et al., 2018; Wang et al., 2020). Gao et al. (2016) summarized the general conclusion that haze events were mainly caused by high emissions of air pollutants and unfavorable weather conditions in North China Plain (NCP) by analyzing a simulated pollution episode from WRF-Chem during 14-24 January 2010. According to the results from WRF-Chem, Qiu et al. (2017) reported that the direct radiative effects of scattering aerosols were greater than that of absorbing aerosols in NCP during 21-27 February 2014. Gao et al. (2018) found that the interactions between black carbon and planetary boundary layer (PBL) could influence the surface $O_3$ concentration in Nanjing during 17 October 2015 by using the process analysis in WRF-Chem.

According to the reviewer's comments, another two complex air pollution episodes (8-13 July 2015 and 5-11 June 2016) in this region are also selected to conduct simulations for generating general conclusions.

[Figure]

**Figure R1.** The changes in surface-layer ozone due to (a) aerosol-photolysis interaction (API), (b) aerosol-radiation feedback (ARF), and (c) the combined effects (ALL, defined as API+ARF) in the daytime (08:00-17:00 LST) during 28 July to 3 August 2014 (Episode1), 8-13 July 2015 (Episode2) and 5-11 June 2016 (Episode3). The region sandwiched between two black lines is defined as the complex air pollution areas (CAPAs) where the mean daily

PM$_{2.5}$ and MDA8 O$_3$ concentrations in BASE case are larger than 75 µg m$^{-3}$ and 80 ppb. The calculated mean changes (percentage changes) avaraged over CAPAs are also shown at the top of each panel.

Simulated air pollutants (PM$_{2.5}$ and O$_3$) and meteorological variables (T$_2$, RH$_2$, WS$_{10}$, and J[NO$_2$]) during 28 July to 3 August 2014 (Episode1), 8-13 July 2015 (Episode2) and 5-11 June 2016 (Episode3) are compared with observations. In general, both the observed meteorological parameters and pollutant concentrations can be reasonably reproduced by the model, with correlation coefficients (R) of 0.56~0.98 and normalized mean bias (NMB) of –12.0%~+33.4%. More details about the model evaluation are listed in the section 3 in the revised manuscript (**Page 7-8, Line 181-212**). The impacts of aerosol radiation effects on meteorological variables can be found in section 4.1 and 4.2 in the revised manuscript during these three episodes (**Page 9-10, Line 223-254**).

As shown in Fig. R1(a1-a3), API alone leads to overall surface O$_3$ decreases over the entire domain with average reductions of 8.5 ppb (10.1%), 9.0 ppb (10.6%) and 8.3 ppb (10.4%) over CAPAs in the three episodes, respectively. The changes can be explained by the substantially diminished UV radiation due to aerosol loading, which significantly weakens the efficiency of photochemical reactions and restrains O$_3$ formation. However, the decreased surface O$_3$ concentrations due to ARF are only 2.9 ppb (3.1%, Fig. R1(b1)), 1.0 ppb (1.2%, Fig. R1(b2)) and 1.0 ppb (1.1%, Fig. R1(b3)) for the three episodes, which indicates that API is the dominant way for O$_3$ reduction related to aerosol-radiation interactions. Fig. R1(c1-c3) presents the combined effects of API and ARF. Generally, aerosol-radiation interactions decrease the surface O$_3$ concentrations by 11.4 ppb (13.5%), 10.0 ppb (11.9%) and 9.3 ppb (11.6%) averaged over CAPAs in the three episodes, respectively. (**Page 10, Line 256-269**)

2. *Given that government controls have substantially changed emissions in the last decade and will continue into future, how will this research remain relevant in the future or how relevant is it to today's air pollution in China, since the period examined is 7 years ago?*

**Response:**

The stringent Air Pollution Action Plan has been released by the Chinese government in September 2013 to improve the PM$_{2.5}$ air quality. Although the concentrations of PM$_{2.5}$ are decreasing, the concentrations of PM$_{2.5}$ still exceed 35 µg m$^{-3}$, and the O$_3$ levels have continued to increase (Dai et al., 2021). Many studies have found that the decreased PM$_{2.5}$ can be one of the important causes leading to the increase in O$_3$ (Li et al. 2019; Shao et al., 2021). Li et al. (2019) pointed out that the concentrations of PM$_{2.5}$ were decreased by 40% in North China Plain from 2013 to 2017, which reduced the sink of HO$_2$ on aerosol surfaces and resulted in the increase in O$_3$ by analyzing simulation results from the GEOS-Chem model. Meanwhile, the concentrations of O$_3$ can also be influenced by aerosol-radiation interactions, including aerosol-photolysis interaction and aerosol-radiation feedback, which have not been systematically analyzed. The quantification of the impacts of aerosols on O$_3$ is important to well understand the co-benefits associated with reductions in both aerosols and O$_3$.

In this study, we investigate the impacts of aerosol-radiation interactions on surface O$_3$, and find that the combined impacts of weakened photolysis rates and changed meteorological conditions reduce surface-layer O$_3$ concentrations by up to 9.3~11.4 ppb. The result can imply that the decreases in $PM_{2.5}$ can lead to the increase in $O_3$ due to the weakened aerosol-radiation interactions, which indicates that if the government controls the anthropogenic emissions in future by using the same strategy, higher $O_3$ will be observed. The result can further emphasize the importance of tighter controls in $O_3$ precursors (e.g., VOCs) to counteract the increased $O_3$ caused by weakened aerosol-radiation interactions. Therefore, the contributions of different mitigation strategies with the impacts of aerosol-radiation interactions to $O_3$ air quality will be discussed detailedly in our future work.

3. *Is the focus of this research the method in which API and ARF are investigated or the impact of API and ARF in North China? If it is the former than the authors need to reword the abstract, conclusions, and objectives to make it clear that this study is a "proof-of-concept" study on how to best investigate API and ARF in high $O_3$ and $PM_{2.5}$ episodes. If the focus is the latter, the authors need to do additional simulations of other high multi-pollutant episodes, perhaps some closer to current conditions and others in the mid 2000s to see if there is change over time or to make the analysis and conclusions more robust.*

**Response:**

This study mainly focuses on the impacts of API and ARF in North China. According to the reviewer's comments, another two complex air pollution episodes (8-13 July 2015 and 5-11 June 2016) in this region are also selected to conduct simulations for generating general conclusions. The impacts of API and ARF on $O_3$ are shown in Fig. R1, and API is the dominant factor for $O_3$ reduction related to aerosol-radiation interactions in these three episodes.

4. *Does this version of WRF-Chem's CBM-Z and MOSAIC modules have a volatility basis set (VBS) option to simulate secondary organic aerosols and if so is it used? Given that, this is a high $O_3$ and $PM_{2.5}$ episode there should be a substantial amount of secondary organic aerosol from abundant oxidants and precursors that may be missed in the model without an advanced SOA scheme. How do the author's address the impact of SOA on their conclusions?*

**Response:**

The selected gas-phase chemical mechanism (CBM-Z) and the aerosol model (MOSAIC) in this study do not consider the impacts of secondary organic aerosols (SOA). The same schemes have been widely used in many other studies, which mainly focus on the impacts of aerosol-radiation interactions on air pollutants in North China (Ding et al., 2016; Gao et al., 2016; Qiu et al., 2017; Chen et al., 2019; Zhou et al, 2019; Gao et al., 2020).

Thanks for the reviewer's suggestion, and we will consider the impacts of SOA in our future works. A discussion about the impacts of SOA has been added in the revised manuscript as follows: "Gao et al. (2017) added some SOA formation mechanisms into the MOSAIC module by using the volatility basis set (VBS) in WRF-Chem and found that the surface $PM_{2.5}$ concentrations in urban Beijing were reduced by 1.9 µg m$^{-3}$ due to the weakened ARF effect during Asia-Pacific Economic Cooperation (APEC). Similar magnitude can also be found in Zhou et al. (2019) (-1.8 µg m$^{-3}$) who did not consider the impacts of SOA in WRF-Chem when analyzing the impacts of weakened ARF on $PM_{2.5}$ during APEC. Therefore, more work should be conducted to explore the impacts of ARF on $PM_{2.5}$ and $O_3$ concentrations under consideration of SOA in future." (**Page 14-15, Line 391-401**)

5. *The authors are investigating aerosol radiation interactions, but the authors do not evaluate the model's performance against either radiation balance datasets or aerosol optical depth. Since these parameters are more important than surface evaluations of air pollutants to understanding API and ARF, the authors should evaluate their model configuration against satellite AOD and radiation variables such as MODIS or CERES-EBAF.*

**Response:**

[Figure]

**Figure R2.** Comparison of observed and simulated AOD at 550 nm in Beijing (39.98°N, 116.38°E). The observed AOD during the three episodes are collected from AERONET.

Previous studies found that MODIS retrievals tended to overestimate AOD in the NCP during polluted events compared with AERONET AOD (Gao et al., 2015; Li et al., 2016). Therefore, we mainly focus on the comparisons between simulated AOD values and AERONET observations in this work. Figure R2 shows the correlation between observed and simulated AOD at 550 nm in Beijing. In the WRF-Chem model, the AOD at 550 nm are calculated by using the values at 400 and 600 nm according to the Angstrom exponent. Analyzing Fig. R2, the model can reproduce the observed AOD with R of 0.7 and NMB of 7.9%.

According to the reviewer's suggestion, the description of the model evaluation between observed and simulated AOD is added in the revised manuscript (**Page 8, Line 190-193**), and Figure R2 is also added in the supporting information (**Figure S1**).

6. *Are there only three meteorological observation stations in the domain against? If so, why do the authors not also validate their meteorological performance against gridded products like the Climate Research Unit (CRU) datasets to ensure their performance statistics are robust?*

**Response:**

Thanks to the reviewer's comments. More meteorological observations in the analyzed domain (Table R1) have been used to validate the model results, and the locations of each site are shown in Fig. R3.

Figure R4 shows the time series of observed and simulated $T_2$, $RH_2$, $WS_{10}$ and $J[NO_2]$ during the three episodes. The observed $T_2$, $RH_2$, $WS_{10}$ are averaged over the ten meteorological observation stations, and the $J[NO_2]$ are measured at Peking University. Most of the monitored $J[NO_2]$ in Episode3 are unavailable, so the comparison of $J[NO_2]$ in Episode3 is not shown. Generally, the model can depict the temporal variations of $T_2$ fairly well with R of 0.98 and the mean bias (MB) of -1.9~-0.9 °C. For $RH_2$, the R and MB are 0.91~0.97 and -4.0%~1.9%, respectively. Although WRF-Chem model overestimates $WS_{10}$ with the MB of 0.6~0.9 m s$^{-1}$, the R for $WS_{10}$ is 0.70~0.89 and the root-mean-square error (RMSE) is 0.9~1.5 m s$^{-1}$, which is smaller than the threshold of model performance criteria (2 m s$^{-1}$) proposed by Emery et al. (2001). The positive bias in wind speed can also be reproduced in other studies (Zhang et al., 2010; Gao et al., 2015; Liao et al., 2015; Qiu et al., 2017). The predicted $J[NO_2]$ agrees well with the observations with R of 0.97~0.98 and NMB of 6.8%~6.9%.

According to the reviewer's comments, we have modified the model evaluation in the revised manuscript. (**Page 8, Line 195-208**)

The gridded products like the Climate Research Unit (CRU) datasets covers a large area and a longtime period, which aims to improve scientific understanding of the climate system and its interactions with society. However, the spatial (0.5° × 0.5°) and temporal (monthly) resolution may be too coarse to validate the model performance for generating robust results.

**Table R1.** Locations of the ten stations from NOAA's National Climatic Data Center used in this study.

| Station | Latitude (°) | Longitude (°) |
|---|---|---|
| Yuxian | 39.833 | 114.567 |
| Fengning | 41.2 | 116.633 |
| Zhangjiakou | 40.783 | 114.883 |
| Huailai | 40.417 | 115.5 |
| Chengde | 40.967 | 117.917 |
| Beijing | 40.08 | 116.585 |
| Tianjin | 39.1 | 117.167 |
| Binhai | 39.124 | 117.346 |
| Tangshan | 39.65 | 118.1 |
| Baoding | 38.733 | 115.483 |

[Figure]

**Figure R3.** Map of the two WRF-Chem modeling domains with the locations of meteorological (white dots) and environmental (red crosses) observation sites used for model evaluation.

[Figure]

**Figure R4.** Time series of 3-hourly observed (blue dots) and hourly simulated (red lines) (a) 2-m temperature ($T_2$), (b) 2-m relative humidity ($RH_2$), (c) wind speed at 10 m ($WS_{10}$) averaged over ten meteorological observation stations, and (d) surface photolysis rate of $NO_2$ ($J[NO_2]$) during 28 July to 3 August 2014 (Episode1, a1-d1), 8-13 July 2015 (Episode2, a2-d2) and 5-11 June 2016 (Episode3, a3-c3). The calculated correlation coefficient (R), mean bias (MB), normalized mean bias (NMB) and root-mean-square error (RMSE) are also shown.

7. *Given that interactions between $O_3$ and $PM_{2.5}$ are non-linear, how do the authors justify using a simple ratio value (i.e., ROP) to relate these interactions? If this ratio*

*does not account for non-linearity, how useful is this value?*

**Response:**

Thanks to the reviewer's suggestion. As the relationship between $O_3$ and $PM_{2.5}$ is non-linear, and the simple index of ROP can not fully represent the impacts of aerosols on surface $O_3$, so we delete the ROP in the revised manuscript.

8. *The axis labels and legends of Figure 7 are difficult to read. Either each panel should be larger overall or the font sizes of the axes and legends need increased.*

**Response:**

According to the reviewer's suggestion, we have modified the axis labels and legends of Figure 7 and the other figures in the revised manuscript. (**Page 32**)

**Minor Comments:**

1) *In the abstract, there is no context for the values listed. Further reading into the manuscript reveals that these values are the averages in the areas of the complex air pollution areas. The authors should briefly state that these values are for daytime average changes in complex air pollution areas in the abstract. I would also suggest adding a more processed based explanation of the changes in atmospheric state rather than simply listing a long series of values. For example, the authors could state something similar to the following: "Aerosol radiation interactions lead to shortwave dimming at the earth's surface of X, which reduce photolysis rates by X. The dimming stabilizes the atmosphere via surface cooling of X, which reduces PBL height by X. The stabilized atmosphere increases saturation in the lower atmosphere by X. etc…."*

**Response:**

According to the reviewer's suggestion, we have revised the explanation in the abstract as follows: "Our results show that aerosol-radiation interactions decreased the daytime shortwave radiation at surface by 92.4~100.3 W $m^{-2}$ averaged over the complex air pollution areas in these three episodes. The dimming effect reduced the near-surface photolysis rates of $J[NO_2]$ and $J[O^1D]$ by $1.8 \times 10^{-3}$~$2.0 \times 10^{-3}$ $s^{-1}$ and $5.7 \times 10^{-6}$~$6.3 \times 10^{-6}$ $s^{-1}$, respectively. However, the daytime shortwave radiation in the atmosphere was increased by 72.8~85.2 W $m^{-2}$, which made the atmosphere more stable. The stabilized atmosphere decreased the planetary boundary layer height and 10 m wind speed by 129.0~249.0 m and 0.05~0.12 m $s^{-1}$, respectively." (**Page 2, Line 25-33**)

2) *Make it clear throughout the manuscript when you are referring to surface level $O_3$ and $PM_{2.5}$.*

**Response:**

According to the reviewer's suggestion, we have revised the expressions in the whole manuscript.

3) *Lines 179-181: The missing $PM_{2.5}$ could also be from missing SOA formation pathways, as mentioned above, if no advanced SOA formulations are used.*

**Response:**

Thanks for your suggestion. The selected aerosol model (MOSAIC) in this study does not consider the impacts of secondary organic aerosols (SOA). We have deleted this sentence in the revised manuscript.

4) *Is "downward shortwave radiation in the atmosphere" the SWDNT variable from WRF-Chem? If so, the name of this variable is "downward shortwave radiation at the top of the atmosphere".*

**Response:**

Thanks for your comments. In the WRF-Chem model, SWDNT (SWUPT) means the download (upward) shortwave radiation at the top of atmosphere, and SWDNB (SWUPB) represents the download (upward) shortwave radiation at the surface. According to Zhao et al. (2011), the shortwave radiation in the atmosphere (ATM_SW) can be calculated as the difference between TOP_SW (the net shortwave radiation at the top of atmosphere, i.e., SWDNT minus SWUPT) and BOT_SW (the net shortwave radiation at the surface, i.e., SWDNB minus SWUPB).

According to the reviewer's suggestion, we have changed the expressions of BOT_SW (shortwave radiation at the surface) and ATM_SW (shortwave radiation in the atmosphere) in the whole revised manuscript.

5) *Lines 217-218: If ATM_SW is the SWDNT variable, what is causing it to increase? SWDNT is usually controlled by the solar constant. Is it possible this is reflected upward shortwave (SWUPT)?*

**Response:**

ATM_SW represents the shortwave radiation in the atmosphere, and it can be calculated by the following equation: ATM_SW = (SWDNT - SWUPT) – (SWDNB - SWUPB).

6) *Lines 248-249: This should be revised to make it clearer that ARF primarily impacts $O_3$ through changing the NOx distribution.*

**Response:**

According to the comments of Reviewer#1, we have deleted this sentence.

7) *Lines 270-281: Is VMIX increasing surface $O_3$ because it is mixing down higher $O_3$ concentrations from aloft or because vertical mixing is suppressed due to a stable atmosphere?*

**Response:**

VMIX increases the surface $O_3$ concentrations by transporting the higher $O_3$ from aloft to the surface layer. Similar results can also be found in previous studies (Tang et al., 2017; Xing et al., 2017; Gao et al., 2018).

8) *Lines 282-294: Why does the VMIX contribution increase because of API?*

**Response:**

Analyzing the vertical profiles of the differences in contributions from each physical/chemical process to hourly $O_3$ variations caused by API in Fig. 8(b), we found that the contribution of VMIX_DIF is negative in the aloft (among the $9^{th}$ and the $13^{th}$ layers), while it turns to be positive at the lower seven layers, and the positive contribution increases as the height decreases. The positive variation in VMIX due to API may be associated with the different vertical gradient of $O_3$

between BASE and NOAPI cases.

Similar results can also be found in Gao et al. (2020), who concluded that the increased vertical gradients of $O_3$ due to API could enhance the vertical entrainment.

9) *Lines 295-301: Explain why VMIX_DIF and CHEM_DIF are positive during the day due to ARF.*

**Response:**

When the impacts of ARF are considered, PBLH is decreased over CAPAs (Fig. S4(a3-c3)), which indicates that the suppressed PBL in NOAPI restrains the vertical turbulence and prevents $O_3$ being transported from aloft to surface, resulting in lower $O_3$ concentrations at surface when comparing with the simulation results of NOALL. However, as the evolution in boundary layer during the daytime, more $O_3$ can be diffused from the upper layers to the surface in NOAPI, and the differences in hourly variation in surface $O_3$ due to vertical mixing between NOAPI and NOALL are positive. Similar results can also be found in Gao et al. (2018).

The typical VOCs/$NO_x$ ratio is calculated to classify sensitivity regimes and to indicate the possible $O_3$ responses to changes in VOCs and/or $NO_x$ concentrations. $O_3$ production is VOC-limited if the ratio is less than 4, and it is $NO_x$-limited if the ratio is larger than 15 (Edson et al., 2017; Li et al., 2017). The ratio of VOCs/$NO_x$ ranging around 4-15 indicates a transitional regime, where ozone is nearly equally sensitive to each species (Sillman, 1999). As shown in Fig R5(a-f), $O_3$ are mainly formed under the VOC-limited and the transition regimes in CAPAs, which means that the increased concentrations of VOCs and $NO_x$ are favorable for ozone chemical production. As shown in Fig. R5(g-i) and (j-l), both the surface concentrations of VOCs and $NO_x$ are increased when the impacts of ARF are considered. Thus, the contribution of CHEM in NOAPI is larger than that in NOALL. Similar results can also be found in Gao et al. (2018).

[Figure]

**Figure R5.** The ratios of VOCs/$NO_x$ calculated from (a-c) NOALL, and (d-f) NOAPI. The changed surface-layer concentrations of (g-i) VOCs and (j-l) $NO_x$ ($NO_2$+NO, ppb) caused by ARF during the daytime (08:00-17:00 LST) from Episode1 to Episode3. The calculated values averaged over CAPAs are also shown at the top of each panel.

*10) Lines 315-316: Explain how different vertical $O_3$ gradients can cause this change.*
**Response:**

Since the VMIX is closely dependent on atmospheric turbulence and vertical gradients of $O_3$ concentration. The API will increase vertical gradients of $O_3$ to enhance the vertical entrainment (Gao et al., 2020).

**Line Comments:**

*1) Line 49: This should be "Earth's radiative balance" or "Earth's energy balance"*
**Response:**
  Thanks for your suggestion. We have changed the expression in the revised manuscript. (**Page 3, Line 49**)

*2) Lines 54-56: Are these studies all focused on China? If so, state that in the sentence. Change "were" to "are".*
**Response:**
  According to the reviewer's suggestion, we have changed the expression in the revised manuscript. (**Page 3, Line 55**)

*3) Lines 56-63: State the domain and time period of Gao et al., (2015) at the beginning of this statement rather than the end*
**Response:**
  According to the reviewer's suggestion, we have changed the expression in the revised manuscript. (**Page 3, Line 56-63**)

*4) Line 66: Add "the" before North China Plain*
**Response:**
  Thanks for your suggestion. We have added the "the" before North China Plain in the revised manuscript. (**Page 3, Line 66**)

*5) Lines 66-67: If this is referring to surface $PM_{2.5}$ concentrations, add "surface" before $PM_{2.5}$ concentrations.*
**Response:**
  Thanks for your suggestion. We have added the "surface" before $PM_{2.5}$ concentrations in the revised manuscript. (**Page 3, Line 66**)

*6) Line 204: should be "attention"*
**Response:**
  Thanks for your suggestion. We have changed the expression in the revised manuscript. (**Page 9, Line 216**)

*7) Line 256: Center align the equation.*
**Response:**
  This equation has been deleted.

*8) Line: 259: Why are there parentheses in the units?*
**Response:**
  This sentence has been deleted.

9) *Lines 288-289: This sentence is a little confusing. Is Net_DIF the sum of CHEM_DIF, VMIX_DIF, and ADV_DIF? If so, state that explicitly and then indicate what Net_DIF describes.*

**Response:**

Thanks for your suggestion. We have defined the NET_DIF in the revised manuscript. (**Page 11, Line 297**)

10) *Line 321: Remove "in the"*

**Response:**

According to the reviewer's suggestion, we have deleted it in the revised manuscript.

11) *Line 361: Remove "the contribution from VMIX and"*

**Response:**

According to the reviewer's suggestion, we have deleted it in the revised manuscript.

12) *Line 373: Either "A recent study" or "Recent studies have"*

**Response:**

According to the reviewer's suggestion, we have changed the expression in the revised manuscript. (**Page 14, Line 385**)

[revised manuscript text omitted]

---

## Author Response (AR3)

**Response to Comments of Editor**

**(Comments in *italics*)**

**Manuscript number:** acp-2021-119

**Title:** Impacts of aerosol-photolysis interaction and aerosol-radiation feedback on surface-layer ozone in North China during a multi-pollutant air pollution episode

*Please respond to the comments of the reviewer. In particular, it seems that the paper is still not communicating clearly why these 3 high pollution episodes are relevant or important to study, and consequently, how to interpret the conclusions to make broader inferences about other time periods.*

*To me, analyzing 3 short episodes in one location is only sufficient for a paper in ACP if the authors motivate well why those episodes are important to study, putting them in a broader context.*

**Response:**

Thanks for the Editor's helpful comments to this manuscript. We have carefully addressed the Reviewer's comments.

In this study, we aim to quantify the impacts of aerosol-radiation interactions on $O_3$ through two processes (aerosol-photolysis interaction (API) and aerosol-radiation feedback (ARF)) during the complex air pollution events with high concentrations of both $PM_{2.5}$ and $O_3$. These events were selected to obtain the strongest signals of API and ARF. These events were selected also considering that the measurements of $J[NO_2]$ during 2014 to 2015 from Peking University site (Wang et al., 2019) can help to constrain the simulated photolysis rates of $NO_2$. The new findings from our work are that API and ARF both have an effect of reducing $O_3$, but the role of API is much larger than that of ARF during all the simulated episodes, which are important for understanding summertime $O_3$ pollution in China.

**Response:**

We now use the same y-axis in Figures 2 and 3. (**Pages 27 and 28**)

**Reference:**

[revised manuscript text omitted]

---

## Author Response (AR4)

**Response to Comments of Editor**

**(Comments in *italics*)**

**Manuscript number:** acp-2021-119

**Title:** Impacts of aerosol-photolysis interaction and aerosol-radiation feedback on surface-layer ozone in North China during a multi-pollutant air pollution episode

*The last draft was marked "major revision" but it appears to me that no major revision was made. In fact, no changes were made at all to the Introduction and Methods sections. The major comment from the reviewer, which I reiterated, was that the paper does not sufficiently motivate the focus on 3 episodes, nor discuss the relevance of the results for other periods of time. The paper still does not state why these 3 episodes were chosen. The paper also seems to be motivated to study high pollution conditions, but that is not stated in the major objectives. And while the authors responded to my comments and the reviewer's first comment, no changes were made in the paper itself to clarify these points.*

*What is the goal of the paper - to consider high air pollution conditions, and for what purpose?*

*Why were the three episodes chosen, and what do we hope to learn more generally for other conditions?*

*How do the conclusions based on the three episodes relate with other conditions during the years studied (2014-16) and to what extent are they relevant for other years before or after?*

**Response:**

Thanks for the editor's valuable comments. Following your suggestions, substantial revision has been made in the revised paper. The newly updated responses and the final revised manuscript can clearly show how the manuscript is improved.

**Response to "***What is the goal of the paper - to consider high air pollution conditions, and for what purpose?***"**

We have **added** the following paragraphs in the **Introduction** section to clarify the goal of this work and why we consider high pollution events:

"The characteristics of air pollution in China during recent years are changing from the single pollutant (e.g., $PM_{2.5}$, particulate matter with an aerodynamic equivalent diameter of 2.5 μm or less) to multiple pollutants (e.g., $PM_{2.5}$ and ozone ($O_3$)) (Zhao et al., 2018; Zhu et al., 2019), and the synchronous occurrence of high $PM_{2.5}$ and $O_3$ concentrations has been frequently observed, especially during the warm seasons (Dai et al., 2021; Qin et al., 2021). Qin et al. (2021) reported that the co-occurrence of $PM_{2.5}$ and $O_3$ pollution days (days with $PM_{2.5}$ concentration > 75 μg m$^{-3}$ as well as maximum daily 8 h average ozone concentration > 80 ppb) exceeded 324 days in eastern China during 2015-2019. Understanding the complex air pollution is essential for making plans to improve air quality in China."

"The present study aims to quantify the respective/combined impacts of ARF and API on surface $O_3$ concentrations by using the WRF-Chem model, and to identify the prominent physical and/or chemical processes responsible for ARF and API effects by using an integrated process rate

(IPR) analysis embedded in the WRF-Chem model."

**Response to "*Why were the three episodes chosen, and what do we hope to learn more generally for other conditions?*"**

We had three episodes in the previous version of the manuscript. In order to cover different years, a new complex air pollution episode (Episode4, 28 June to 3 July 2017) is **added** in the revised manuscript. As shown in Fig. 6 (see below) of the revised manuscript, all the episodes show same conclusion that the reduction in $O_3$ by API is larger than that by ARF.

To clarify the reasons of choosing these four episodes, we have **added** the following paragraph in the **Introduction** section:

"We carry on simulations and analyses on four multi-pollutant air pollution episodes (Episode1: 28 July to 3 August 2014; Episode2: 8-13 July 2015; Episode3: 5-11 June 2016; Episode4, 28 June to 3 July 2017) in North China with high $O_3$ and $PM_{2.5}$ levels (the daily mean $PM_{2.5}$ and the maximum daily 8-h average $O_3$ concentration are larger than 75 µg m$^{-3}$ and 80 ppb, respectively). These episodes are selected because (1) these events with high concentrations of both $PM_{2.5}$ and $O_3$ are the major subjects of air pollution control, (2) high concentrations of both $PM_{2.5}$ and $O_3$ allow one to obtain the strongest signals of ARF and API, (3) the measurements of $J[NO_2]$ during 2014 and 2015 from Peking University site (Wang et al., 2019) can help to constrain the simulated photolysis rates of $NO_2$, and (4) selected events cover different years of 2014 to 2017 during which the governmental Air Pollution Prevention and Control Action Plan was implemented (the changes in emissions and observed $PM_{2.5}$ in the studied region during 2014-2017 are shown in Fig. S1). We expect that the conclusions obtained from multiple episodes represent the general understanding of the impacts of ARF and API."

[Figure]

**Figure 6.** The changes in surface-layer $O_3$ due to (a) aerosol-photolysis interaction (API), (b) aerosol-radiation feedback (ARF), and (c) the combined effects (ALL, API+ARF) in the daytime (08:00-17:00 LST) during 28 July to 3 August 2014 (Episode1), 8-13 July 2015 (Episode2), 5-11 June 2016 (Episode3) and 28 June to 3 July 2017 (Episode4). The regions sandwiched between two black lines are defined as the complex air pollution areas (CAPAs). The changes (percentage changes) in $O_3$ concentrations caused by API, ARF, and ALL avaraged over CAPAs are shown at the top of each panel.

[Figure]

**Figure S1.** Trends of emissions over 2014–2017 from MEIC emission inventory and the observed annual mean PM$_{2.5}$ concentrations in the studied domain during 2014-2017.

**Response to *"How do the conclusions based on the three episodes relate with other conditions during the years studied (2014-16) and to what extent are they relevant for other years before or after?"***

**(1)** *How do the conclusions based on the three episodes relate with other conditions during the years studied?*

To address the editor's comments, three additional episodes with different conditions of air pollution, i.e., (1) $PM_{2.5}$ pollution alone (Episode_add1, the daily mean $PM_{2.5}$ concentration is larger than 75 µg m$^{-3}$), (2) neither $PM_{2.5}$ nor $O_3$ exceed air quality standard (Episode_add2, the daily mean $PM_{2.5}$ and maximum daily 8-h average $O_3$ concentration are smaller than 75 µg m$^{-3}$ and 80 ppb, respectively), and (3) $O_3$ pollution alone (Episode_add3, the maximum daily 8-h average $O_3$ concentration is larger than 80 ppb) are simulated to examine the impacts of API and ARF on $O_3$. Detailed information about these three additional episodes is summarized in Table S3 (see below) of the revised manuscript.

**Table S3.** Three additional episodes with different levels of $PM_{2.5}$ and $O_3$.

| Case | Time | $PM_{2.5}$ pollution (concentration) | $O_3$ pollution (concentration) |
|---|---|---|---|
| **Episode_add1** | 2014.10.7-2014.10.11 | √ (223.5 µg m$^{-3}$) | × (46.9 ppb) |
| **Episode_add2** | 2016.6.13-2016.6.17 | × (36.5 µg m$^{-3}$) | × (62.4 ppb) |
| **Episode_add3** | 2017.6.15-2017.6.20 | × (61.9 µg m$^{-3}$) | √ (103.6 ppb) |

In Episode_add1, Episode_add2 and Episode_add3, API alone is simulated to reduce surface $O_3$ averaged over each episode and over the entire simulated domain (38.01~41.45 °N, 114.52~118.28 °E) by 15.3 ppb (29.3%), 4.4 ppb (6.8%) and 4.5 ppb (5.3%), respectively, and ARF alone reduces surface $O_3$ by 3.9 ppb (6.2%), 0.6 ppb (1.0%), and 0.1 ppb (0.1%), respectively (see below, Fig. S13 of the revised manuscript). All the results confirm the same conclusion that the reduction in $O_3$ by API is larger than that by ARF.

We have also **added** the following discussion about these three additional episodes in the **Discussion** section:

"We presented above the results from our simulations of multi-pollutant air pollution episodes. In order to show that the conclusion of this work can be applied to other conditions of air pollution, three additional situations are carried out, i.e., (1) $PM_{2.5}$ pollution alone (Episode_add1, the daily mean $PM_{2.5}$ concentration is larger than 75 µg m$^{-3}$), (2) neither $PM_{2.5}$ nor $O_3$ exceed air quality standard (Episode_add2, the daily mean $PM_{2.5}$ and maximum daily 8-h average $O_3$ concentration are smaller than 75 µg m$^{-3}$ and 80 ppb, respectively), and (3) $O_3$ pollution alone (Episode_add3, the maximum daily 8-h average $O_3$ concentration is larger than 80 ppb). Detailed information about these three additional episodes is listed in the supporting information (Text S1 and Table S3). Analyzing Episode_add1, Episode_add2 and Episode_add3 in Fig. S13, API alone is simulated to reduce surface $O_3$ averaged over each episode and over the entire domain by 15.3 ppb (29.3%), 4.4 ppb (6.8%) and 4.5 ppb (5.3%), respectively, and ARF alone reduces surface $O_3$ by 3.9 ppb (6.2%), 0.6 ppb (1.0%), and 0.1 ppb (0.1%), respectively. All the results confirm the same conclusion that the reduction in $O_3$ by API is larger than that by ARF."

**(2)** *How do the conclusions based on the three episodes to what extent are they relevant for other*

*years before or after?*”

Air pollution in China was characterized by high concentrations of $PM_{2.5}$ before 2014 (Li et al., 2019a; Zhang et al., 2019) and by synchronous occurrence of high $PM_{2.5}$ and $O_3$ or high levels of $O_3$ alone after 2017 (Dai et al., 2021; Li et al., 2019b; Li et al., 2020; Qin et al., 2021). Episode_add1, Episode1-Episode4, and Episode_add3 can represent these situations, respectively. Therefore, we believe that our general conclusion can also be applied to the years before 2014 and after 2017.

[Figure]

**Figure S13.** The changes in surface-layer $O_3$ due to (a) aerosol-photolysis interaction (API), (b) aerosol-radiation feedback (ARF), and (c) the combined effects (ALL, API+ARF) in the daytime (08:00-17:00 LST) of 7-11 October 2014 (Episode_add1), 13-17 June 2016 (Episode_add2) and 15-20 June 2017 (Episode_add3). The changes (percentage changes) in $O_3$ concentrations caused by API, ARF and ALL avaraged over the entire simulated domain are also shown at the top of each panel.

**Response:**

It's difficult to carry out simulations of complex air pollution events during 2001-2005, because the national observations of $PM_{2.5}$ and $O_3$ concentrations were not available until 2013. Thus, the time and the area of complex air pollution events cannot be determined for earlier years. To address your concerns, we now have four episodes (Episode1: 28 July to 3 August 2014; Episode2: 8-13 July 2015; Episode3: 5-11 June 2016; Episode4, 28 June to 3 July 2017) in the revised manuscript. These selected events cover different years of 2014 to 2017 during which the governmental Air Pollution Prevention and Control Action Plan was implemented (the changes in emissions and observed $PM_{2.5}$ in the studied region during 2014-2017 are shown in Fig. R1).

To further address yours and editor's comments, three additional situations under different conditions of air pollution, i.e., (1) $PM_{2.5}$ pollution alone (Episode_add1, the daily mean $PM_{2.5}$ concentration is larger than 75 µg m$^{-3}$), (2) neither $PM_{2.5}$ nor $O_3$ exceed air quality standard (Episode_add2, the daily mean $PM_{2.5}$ and maximum daily 8-h average $O_3$ concentration are smaller than 75 µg m$^{-3}$ and 80 ppb, respectively), and (3) $O_3$ pollution alone (Episode_add3, the maximum daily 8-h average $O_3$ concentration is larger than 80 ppb) are simulated to examine the impacts of

API and ARF on $O_3$. All the results confirm the same conclusion that the reduction in $O_3$ by API is larger than that by ARF. (Please see our responses to Editor's comments).

Air pollution in China was characterized by high concentrations of $PM_{2.5}$ before 2014 (Li et al., 2019a; Zhang et al., 2019) and by synchronous occurrence of high $PM_{2.5}$ and $O_3$ or high levels of $O_3$ alone after 2017 (Dai et al., 2021; Li et al., 2019b; Li et al., 2020; Qin et al., 2021). Episode_add1, Episode1-Episode4, and Episode_add3 can represent these situations, respectively. Therefore, we believe that our general conclusion can also be applied to the years before 2014 and after 2017.

[Figure]

**Figure R1.** Trends of emissions over 2014–2017 from MEIC emission inventory and the observed annual mean $PM_{2.5}$ concentrations in the studied domain during 2014-2017.

2. *The authors have added caveats to the conclusion to address the issue of lacking SOA formation pathways in their simulations. These listed caveats are important, but the authors have overlooked the possibility that increased $O_3$ from $PM_{2.5}$ reductions will generate more SOA via increased oxidation. This feedback could partially compensate the increased $O_3$ formation the authors predict will happen.*

**Response:**

Thanks for pointing this out. We agree with your helpful suggestion that the lacking SOA formation pathways in these simulations may underestimate the impacts of aerosol-photolysis interaction (API) and aerosol-radiation feedback (ARF) on surface ozone.

The discussions about the impacts of SOA have been **revised** as follows: "In the current CBMZ and MOSAIC schemes, the formation of SOA (secondary organic aerosol) is not included (Gao et al., 2015; Chen et al., 2019). The absence of SOA can underestimate the impacts of API and ARF on $O_3$. Meanwhile, the lack of SOA may lead to weaker heterogeneous reactions to result in higher $O_3$ concentrations (Li et al., 2019c). The net effect of the two processes will be discussed and quantified in our future study."

**Minor Comments:**

1. The response to the previous Reviewer2 Minor comments 7-10 should be included in the manuscript if not already done to facilitate ease of understanding.

**Response:**

The previous Reviewer 2's Minor Comments 7-10 have been **added** into Section 4.4 of the revised manuscript as follows:

(1) Since VMIX increases the surface $O_3$ concentrations by transporting $O_3$ from aloft (where $O_3$ concentrations are high) to the surface layer (Tang et al., 2017; Xing et al., 2017; Gao et al.,

2018).

(2) The positive change in VMIX due to API may be associated with the different vertical gradient of $O_3$ between BASE and NOAPI cases (Gao et al., 2020), as shown in Fig. 8a.

(3) The positive VMIX_DIF is related to the evolution in boundary layer during the daytime. The VOCs/$NO_x$ ratio is calculated to classify sensitivity regimes and to indicate the possible $O_3$ responses to changes in VOCs and/or $NO_x$ concentrations. $O_3$ production is VOC-limited if the ratio is less than 4, and is $NO_x$-limited if the ratio is larger than 15 (Edson et al., 2017; Li et al., 2017). The ratio of VOCs/$NO_x$ ranging around 4-15 indicates a transitional regime, where ozone is nearly equally sensitive to both species (Sillman, 1999). As shown in Fig. S7, (a-f), $O_3$ is mainly formed under the VOC-limited and the transition regimes in CAPAs. As shown in Figs. S7(g-i) and S7(j-l), both the surface concentrations of VOCs and $NO_x$ are increased when the impacts of ARF are considered. Thus, the contribution of CHEM in NOAPI is larger than that in NOALL.

(4) The positive variation in VMIX due to API may be associated with the different vertical gradient of $O_3$ between BASE and NOAPI again.

2. The color bars for Figures 5 and 6 needs to be the same for all episodes to facilitate easy comparison.

**Response:**

According to the reviewer's suggestion, we now use the same color bars in Figures 5 and 6 in the revised manuscript.

3. In Figures 2 and 3 the y-axis need to be consistent for all episodes to facilitate ease of comparison

**Response:**

We now use the same y-axis in Figures 2 and 3 in the revised manuscript.

**Reference:**

[revised manuscript text omitted]

---

## Author Response (AR5)

**Response to Comments of Editor**

**(Comments in *italics*)**

**Manuscript number:** acp-2021-119

**Title:** Impacts of aerosol-photolysis interaction and aerosol-radiation feedback on surface-layer ozone in North China during a multi-pollutant air pollution episode

*The reviewer makes some constructive suggestions which are generally in line with my earlier comments. The paper has improved from the earliest submissions, and I think that it can still improve with attention to these comments.*

**Response:**

Additional simulations have been conducted following the Reviewer's comments, and the results from these new simulations also support the general conclusion we obtained in the manuscript.

As described below in our point-to-point responses, we have carefully revised the manuscript and addressed the Reviewer's comments.

**Response to Comments of Reviewer #2**

**(Comments in *italics*)**

**Manuscript number:** acp-2021-119

**Title:** Impacts of aerosol-photolysis interaction and aerosol-radiation feedback on surface-layer ozone in North China during a multi-pollutant air pollution episode

*This revised paper is well written and well reasoned but in previous round of review, several key issues were pointed out. Specifically in the last round of review, the editor posed three key questions that the authors needed to address in the manuscript with which I agree.*
*1) What is the goal of the paper - to consider high air pollution conditions, and for what purpose?*
*2) Why were the episodes chosen, and what do we hope to learn more generally for other conditions?*
*3) How do the conclusions based on the CAPA episodes relate with other conditions during the years studied (2014-16) and to what extent are they relevant for other years before or after?*
*I believe that the authors have fully answered question 1 and the clear motivation has improved the manuscript. However, the current revision only partially answers questions 2 and 3. My major and minor comments are as follows:*

**Response:**

Thanks to the Reviewer for the valuable comments and suggestions which are helpful for improving our manuscript. We have revised the manuscript carefully, as described in our point-to-point responses to the comments.

**Major Comments:**

*1a) The authors addressed the questions about other conditions by conducting 3 additional simulations with conditions other than complex air pollution episodes. These simulations are important and reveal some interesting findings. Particularly, the O3 decrease in a high PM episode from API and ARF is close to double the impact in a complex episode and in cases without high PM pollution the impact is close to half the impact of a complex episode. However, the authors treat these cases as an afterthought with them being mentioned only in the last few lines of the conclusion/supplementary material. These cases need to be incorporated into the main text as another results section and they need more descriptive names such as: High PM2.5 (HI_PM), Low Pollution (LOW_POL), and High O3 (HI_O3).*

**Response:**

The main purpose of this study is to quantify the respective/combined impacts of ARF and API on surface $O_3$ concentrations during complex air pollution episodes, as stated in the title of the manuscript. We thank the Reviewer for the positive comment on the motivation of this study.

In order to make the conclusions more general, another six simulations under different air pollution conditions (i.e., $PM_{2.5}$ pollution alone, $O_3$ pollution alone, and neither $PM_{2.5}$ nor $O_3$ exceed air quality standard), as suggested by the Editor and the Reviewer, are conducted. All the results confirm the same conclusion that the reduction in $O_3$ by API is larger than that by ARF.

In the revised manuscript, we have changed the names of these additional simulations following the suggestion of the Reviewer. In order not to distract from the major purpose of the article, we have added a new section of 'Discussions' before the Conclusions to present the results from these additional simulations.

[revised manuscript text omitted]

*1b)* *The authors should also consider possibly simulating an additional case with each of these conditions to see if the differences between them and the CAPAs are robust, or alternatively they should demonstrate that the episodes they selected are representative of those conditions throughout the period of interest (2014-2017).*

**Response:**

Following the Reviewer's suggestion, we have carried out simulation of one more case for each of air pollution conditions (i.e., High_PM_Episode2, High_O$_3$_Episode2, and Low_POL_Episode2). Please see our responses to your major comment 1a) for the results from these new simulations. All the results confirm the same conclusion that the reduction in O$_3$ by API is larger than that by ARF (Table 2).

*2)* *The authors have still not addressed historical conditions before 2014. The authors have stated this is because national observations are not available to pin-point complex air pollution episodes before 2013. However, the authors have also stated "Air pollution in China was characterized by high concentrations of PM2.5 before 2014 (Li et al., 2019a; Zhang et al., 2019) and by synchronous occurrence of high PM2.5 and O3 or high levels of O3 alone after 2017 (Dai et al., 2021; Li et al., 2019b; Li et al., 2020; Qin et al., 2021)." If there are no observations before 2013 to justify this statement, how is this statement supported? At a minimum, the authors should point out the lack of data before their period of interest in the manuscript as a reason they cannot explore the magnitudes of API and ARF before 2014. However, if the authors are at least aware of a high PM2.5 case from the 2001-2005 period it would be worth comparing that with the current HI_PM case to see if the magnitude of the API and ARF impacts have decreased in a significant way.*

**Response to "*If there are no observations before 2013 to justify this statement, how is this statement supported?*"**

The open official national data are not available before 2013 as the China National Environmental Monitoring Center (CNEMC) network was established in 2013 (Chinese State Council, 2013). Before 2013, only the US Embassy had publicly accessible and continuous PM$_{2.5}$ observation data in Beijing. Daily PM$_{2.5}$ from the US Embassy in Beijing is available between 18 February 2009 and 30 June 2017 (http://www.stateair.net/web/historical/1/1.html) and is shown in Fig. R1 below. PM$_{2.5}$ concentrations were very high before 2014 (from 18 February 2009 to 31 December 2013), with 935 days (54%) above 75 μg m$^{-3}$ and 345 days (20%) above 150 μg m$^{-3}$ in Beijing, respectively. However, there were no observations of O$_3$ from the US Embassy, which make it difficult to select the time and location of complex air pollution events and to evaluate the model results before 2013.

The general situation of air pollution from 2013 to 2019 was presented by Li et al. (2020). As shown in Fig. R2 (see below, taken from Li et al. (2020)), since the implementation of the Chinese governmental Air Pollution Action Plan in 2013, the concentrations of PM$_{2.5}$ decreased significantly, whereas the concentrations of O$_3$ increased steadily. Meanwhile, few episodes of complex air pollution events were detected in north China in 2013 due to the high concentrations of PM$_{2.5}$ and low concentrations of O$_3$. The observed annual mean concentrations of PM$_{2.5}$ decreased from 82.0 μg m$^{-3}$ in 2014 to 49.8 μg m$^{-3}$ in 2018 over the North China Plain (NCP) (Chen et al., 2020), which still exceeded the National Ambient Air Quality Standards (35 μg m$^{-3}$). Meanwhile, the summertime mean MDA8 O$_3$ concentrations frequently reached or exceeded the 80 ppb in NCP after 2017 (Fig. R2). That's why we say "Air pollution in China was characterized by high concentrations of PM$_{2.5}$ before 2014, and by synchronous occurrence of high PM$_{2.5}$ and O$_3$ or high levels of O$_3$ alone after 2017".

[Figure]

**Figure R1.** Time series of the observed daily PM$_{2.5}$ concentrations in Beijing from the US Embassy from 18 February 2009 to 30 June 2017. The green and red dash lines represent the concentrations of 75 and 150 μg m$^{-3}$, respectively.

[Figure]

**Figure R2.** Summer (June-July-August) concentrations of maximum MDA8 ozone (a), mean MDA8 ozone (b), maximum $PM_{2.5}$ (c), and mean $PM_{2.5}$ (d) for 2013–2019 at the network operated by the China Ministry of Ecology and Environment (MEE). Rectangles denote the four megacity clusters: the North China Plain (NCP; 34–41 °N, 113–119 °E), the Yangtze River Delta (YRD; 30–33 °N, 119–122 °E), the Pearl River Delta (PRD; 21.5–24 °N, 112–115.5 °E), and the Sichuan Basin (SCB; 28.5–31.5 °N, 103.5–107 °E). (Li et al., 2020)

**Response to** *"At a minimum, the authors should point out the lack of data before their period of interest in the manuscript as a reason they cannot explore the magnitudes of API and ARF before 2014."*

According to the Reviewer's comments, the limitation section is updated in our revised manuscript: "(2) The CNEMC network was built in 2013. Before 2013, the national observations of $PM_{2.5}$ and $O_3$ concentrations were not available, which make it difficult to select the time and location of complex air pollution events and to evaluate the model results. Based on observation data, we were mainly focused on impacts of ARF and API on surface $O_3$ during complex air pollution episodes from 2014 to 2017. Additional simulations of High_PM, High_$O_3$, and Low_POL support the conclusion obtained from the complex air pollution episodes that the reduction in $O_3$ by API is larger than that by ARF."

**Response to** *"However, if the authors are at least aware of a high PM2.5 case from the 2001-2005 period it would be worth comparing that with the current HI_PM case to see if the magnitude of the API and ARF impacts have decreased in a significant way."*
    Thanks to the Reviewer's suggestion. The comparison is limited by the availability of observational data, as we explained above.

**Minor comments:**

*1. Line 78: field not "filed"*

**Response:**
    Changed.

*2. Line 84: Should the unit be μg m-3 or ppb?*

**Response:**
    The unit is μg m$^{-3}$ in Xing et al. (2017) paper.

*3. Line 206: done instead of "did"*

**Response:**
    We have changed the "did" to "done" in the revised manuscript.

*4. Line 291: should be "was caused"*

**Response:**
    Corrected.

*5. Line 414: How can API contribute more than 100% to a reduction in O3? The word contribution implies "percent of" not "percent change". If these values are a relative difference (i.e., percent change) then a word other than contribute needs to be used.*

**Response:**
    We have revised the sentence as follows: "The changed meteorological variables and weakened photochemistry reaction further reduced surface-layer $O_3$ concentration by 10.0~11.4 ppb, with relative changes of 74.6%~106.5% by API and of -6.5%~25.4% by ARF."

*6. Line 430: during "the" warm season*

**Response:**

Corrected.

[revised manuscript text omitted]